# Small molecules restore mutant mitochondrial DNA polymerase activity

Sebastian Valenzuela[1], Xuefeng Zhu[1], Bertil Macao[1], Mattias Stamgren[2], Carol Geukens[2], Paul S. Charifson[3], Gunther Kern[3], Emily Hoberg[1], Louise Jenninger[1], Anja V. Gruszczyk[2], Seoeun Lee[1], Katarina A. S. Johansson[1], Javier Miralles Fusté[2], Yonghong Shi[2], S. Jordan Kerns[3], Laleh Arabanian[2], Gabriel Martinez Botella[3], Sofie Ekström[2], Jeremy Green[3], Andrew M. Griffin[3], Carlos Pardo-Hernández[2], Thomas A. Keating[3], Barbara Küppers-Munther[2], Nils-Göran Larsson[4], Cindy Phan[2], Viktor Posse[2], Juli E. Jones[3], Xie Xie[2], Simon Giroux[3✉], Claes M. Gustafsson[1✉] & Maria Falkenberg[1✉]

Mammalian mitochondrial DNA (mtDNA) is replicated by DNA polymerase γ (POLγ), a heterotrimeric complex consisting of a catalytic POLγA subunit and two accessory POLγB subunits[1]. More than 300 mutations in *POLG*, the gene encoding the catalytic subunit, have been linked to severe, progressive conditions with high rates of morbidity and mortality, for which no treatment exists[2]. Here we report on the discovery and characterization of PZL-A, a first-in-class small-molecule activator of mtDNA synthesis that is capable of restoring function to the most common mutant variants of POLγ. PZL-A binds to an allosteric site at the interface between the catalytic POLγA subunit and the proximal POLγB subunit, a region that is unaffected by nearly all disease-causing mutations. The compound restores wild-type-like activity to mutant forms of POLγ in vitro and activates mtDNA synthesis in cells from paediatric patients with lethal *POLG* disease, thereby enhancing biogenesis of the oxidative phosphorylation machinery and cellular respiration. Our work demonstrates that a small molecule can restore function to mutant DNA polymerases, offering a promising avenue for treating *POLG* disorders and other severe conditions linked to depletion of mtDNA.

Mitochondria are central to health and disease, and dysfunctions in mitochondria are linked to cardiovascular diseases, neurodegeneration, metabolic syndrome and cancer[3,4]. Mammalian mtDNA encodes essential protein subunits of the oxidative phosphorylation (OXPHOS) complexes, which are responsible for producing the majority of cellular ATP[1]. *POLG* mutations are a leading cause of inherited mitochondrial disorders, and many different disease-causing mutations have been described[2,5] (https://tools.niehs.nih.gov/polg/). These mutations impair POLγ activity and lead to mtDNA depletion and/or deletions in affected patients[2]. Whereas most mutations are rare, a subset of three amino acid substitutions, A467T, W748S and G848S, has been identified in about 70% of affected patients[2]. Disorders caused by *POLG* mutations encompass a spectrum of overlapping clinical presentations, and the age at which symptoms first appear generally corresponds to the specific clinical features that are observed. Early onset (0–12 years of age) is associated with severe mtDNA depletion and a very short life expectancy, often less than a year. Symptoms include global developmental delay, seizures, hypotonia, muscle weakness and liver dysfunction. In the juvenile or adult-onset form (12–40 years of age), peripheral neuropathy, ataxia, seizures, stroke-like episodes and progressive external ophthalmoplegia are observed. In the late-onset form (after 40 years of age), ptosis and progressive external ophthalmoplegia are predominant, along with peripheral neuropathy, ataxia and muscle weakness. So far, there have been no effective therapeutic strategies available to treat or cure these severe, progressive disorders[6].

## PZL-A stimulates POLγ activity

We set out to identify compounds that could stimulate POLγ activity and mitigate the phenotypes associated with impaired enzyme function. Given the extensive heterogeneity of pathogenic mutations identified in the *POLG* gene, a therapeutic strategy targeting individual mutations is not feasible. Instead, we hypothesized that a molecule capable of enhancing the activity of wild-type POLγ might also be effective across various POLγ mutations. Therefore, we initially screened compounds against wild-type POLγ and subsequently focused our studies on the disease-associated variants of POLγ.

An initial screen of approximately 270,000 compounds led to the discovery of compound **1** (Fig. 1a). When evaluated in a high-throughput recombinant in vitro DNA synthesis assay (Fig. 1b), compound **1** displayed modest half-maximal activity concentration (AC₅₀, the concentration required to achieve 50% of the maximum effect) for wild-type

[1]Department of Medical Biochemistry and Cell Biology, University of Gothenburg, Gothenburg, Sweden. [2]Pretzel Therapeutics, Mölndal, Sweden. [3]Pretzel Therapeutics, Waltham, MA, USA. [4]Department of Medical Biochemistry and Biophysics, Karolinska Institutet, Stockholm, Sweden. ✉e-mail: sgiroux@pretzeltx.com; claes.gustafsson@medkem.gu.se; maria.falkenberg@medkem.gu.se

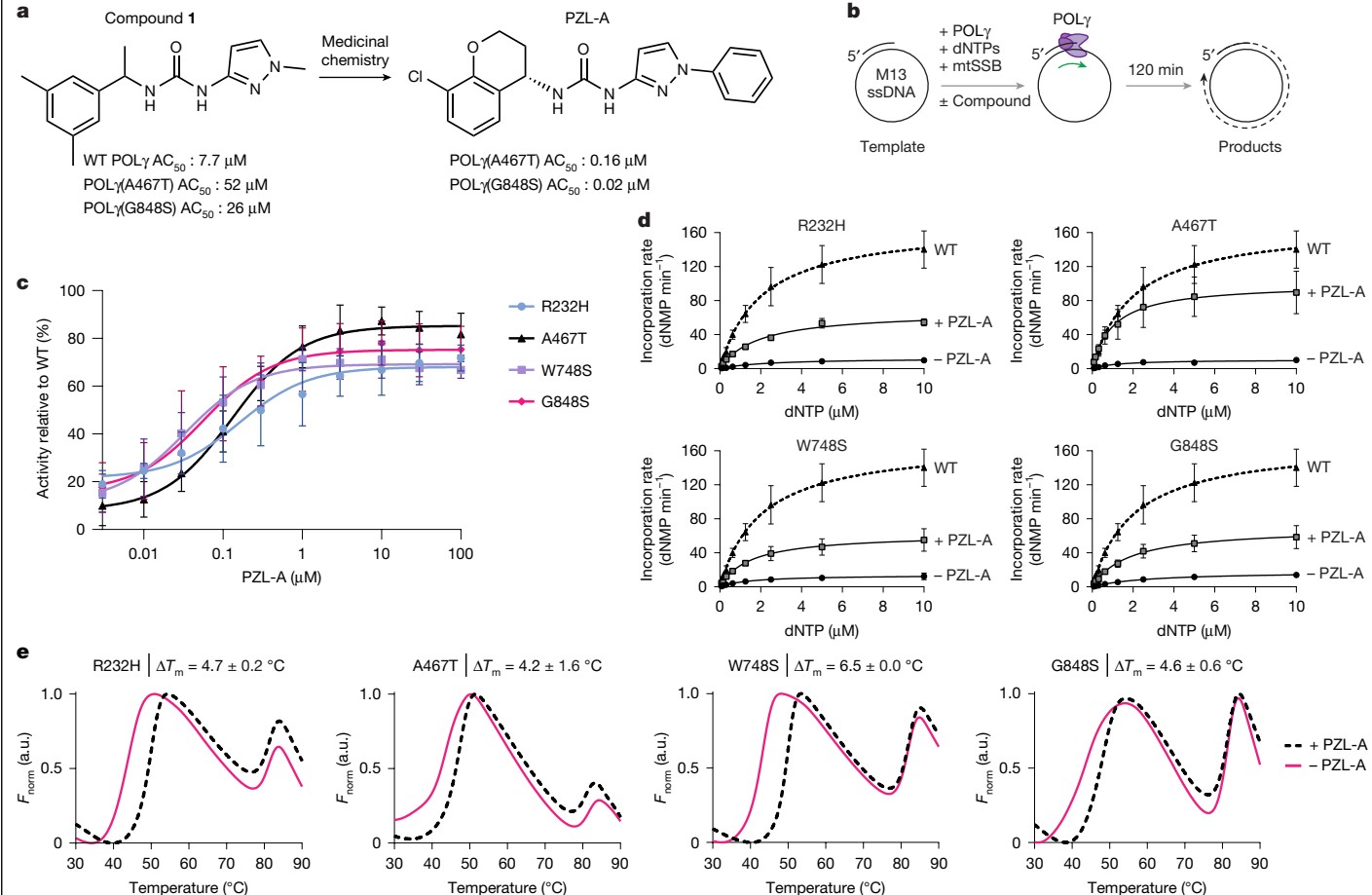

**Fig. 1 | PZL-A activates mutant POLγ. a**, Chemical structure of compound **1**, a high-throughput screening hit, and PZL-A, a more potent compound produced by a medicinal chemistry optimization effort. $AC_{50}$ values are shown for two POLγ mutants and wild-type. **b**, Schematic representation of the primer-elongation assay used for hit-to-lead optimization of POLγ activators. A short 20-nucleotide (nt) primer was annealed to circular, single-stranded M13mp18 DNA, providing a preexisting 3′-OH group from which POLγ could initiate DNA synthesis. POLγ activity was followed by monitoring the increase of SYBR Green fluorescence over time as the dye bound to newly formed dsDNA. Drawing by Jennifer Uhler (copyright holder). **c**, Increasing concentrations of PZL-A activate the indicated mutant POLγ variants in a dose-dependent manner (mean ± s.d.;

$n = 3$ independent experiments). **d**, Steady-state kinetics of dNMP incorporation (mean ± s.d.; WT: $n = 4$; mutants: $n = 3$ independent experiments). Rates were determined by dividing the initial incorporation rates by the concentration of active enzyme–DNA complex (Extended Data Fig. 1b). PZL-A enhances catalytic efficiency in all four mutants. **e**, Representative plots of differential scanning fluorimetry performed with POLγA mutants (R232H, A467T, W748S and G848S) in complex with POLγB, in the absence or presence of 10 μM PZL-A. Values are presented as normalized fluorescence ($F_{norm}$) in arbitrary units (a.u.). The presence of PZL-A induces a shift in melting temperature in all four POLγA mutants and is given as $\Delta T_m$ (mean ± s.d.; $n = 3$ independent experiments). For source data of blots in **d**, see Supplementary Figs. 1 and 2.

POLγ and two mutant forms, A467T and G848S. Compound **1** underwent further hit-to-lead optimization, leading to the more potent PZL-A (Fig. 1a). PZL-A exhibited a robust stimulatory effect on common, disease-causing forms of POLγ, including mutations in the exonuclease domain (R232H), the DNA polymerase domain (A467T and G848S), and the intervening linker region (W748S) of POLγA. Across all these mutations, PZL-A exhibited $AC_{50}$ values in the nanomolar range (20–200 nM) and aided a recovery of enzymatic activity relative to wild-type (Fig. 1c). The stimulatory effect was apparent across a broad range of dNTP concentrations (Extended Data Fig. 1a).

Analysis of kinetic parameters, calculated using the Michaelis–Menten steady-state model, revealed that PZL-A enhances the rate of dNTP incorporation (Fig. 1d, Extended Data Fig. 1b, Table 1). We observed an increase of measured maximum saturated reaction rate ($V_{max}$) and the overall reaction rate constant ($k_{cat}$), with milder effects on the apparent Michaelis constant ($K_{m\_app,\ dNTP}$). In addition to its C-terminal DNA polymerase domain, POLγA also contains an N-terminal 3′–5′ exonuclease domain that is crucial for proofreading during mtDNA synthesis[7–9]. We monitored whether PZL-A could influence exonuclease activity of mutant POLγ variants but observed no negative effects; all

mutants were able to perform proofreading in the presence of PZL-A (Extended Data Fig. 1c–f).

We also confirmed that PZL-A interacts directly with the POLγ holoenzyme by monitoring effects on thermal stability. The compound had a stabilizing effect, increasing the unfolding temperature of all the mutant POLγA variants tested[10] (Fig. 1e).

## PZL-A binds at the POLγA–POLγB interface

To investigate how PZL-A interacts with mutant versions of POLγ during DNA synthesis, we used single-particle cryogenic electron microscopy (cryo-EM) to determine the structures of POLγ in its elongating conformation, both in the absence and in the presence of PZL-A. The elongation complexes were obtained by incubating the enzyme with a primer template DNA substrate, the correct incoming nucleotide (dCTP) and $Ca^{2+}$ ions to halt the polymerase in its elongating state[11]. Reconstructions of two different mutants (A467T and G848S) and wild-type POLγ in complex with template DNA were refined to resolutions of 2.4–2.7 Å using a published crystal structure[12] as a starting point for model building (Extended Data Table 1 and Extended Data Figs. 2–4).

**Table 1 | Steady-state kinetic parameters**

| | | −PZL-A | | | +PZL-A | | |
|---|---|---|---|---|---|---|---|
| | $K_{d\_app,\ PZL-A}$ (nM) | $K_{m\_app,\ dNTP}$ (µM) | $k_{cat}$ (min$^{-1}$) | $V_{max}$ (nM min$^{-1}$) | $K_{m\_app,\ dNTP}$ (µM) | $k_{cat}$ (min$^{-1}$) | $V_{max}$ (nM min$^{-1}$) |
| WT | - | 2.0±0.3 | 170.1±10.2 | 75.9 | - | - | - |
| R232H | 18.7±2.7 | 2.3±0.3 | 12.4±0.6 | 4.9 | 1.9±0.2 | 67.0±2.8 | 30.6 |
| A467T | 57.1±7.4 | 1.2±0.3 | 10.5±0.7 | 3.6 | 1.0±0.3 | 100.0±8.7 | 38.8 |
| W748S | 16.1±1.3 | 1.5±0.4 | 14.0±1.2 | 6.5 | 1.5±0.3 | 62.2±4.4 | 29.2 |
| G848S | 25.7±3.8 | 2.7±0.4 | 18.0±1.2 | 7.1 | 1.8±0.4 | 69.4±5.0 | 31.4 |

Kinetic parameters (apparent $K_m$ for dNTPs ($K_{m\_app,\ dNTP}$) and $k_{cat}$) were extracted from the experiments presented in Fig. 1d (wild-type: $n=4$; mutants: $n=3$). $V_{max}$ was calculated by multiplying $k_{cat}$ by the concentration of active enzyme–DNA complex (Extended Data Fig. 1b). Apparent dissociation constant for PZL-A ($K_{d\_app,\ PZL-A}$) was determined from time-course experiments performed at a fixed dNTP concentration (20 µM) by titrating PZL-A across 8 concentrations (R232H or G848S: $n=3$; A467T or W748S: $n=4$). Data are mean±s.e.m. from 3–4 independent experiments. For source data, see Supplementary Data (Source Data Table 1) and Supplementary Figs. 1–3.

The A467T and G848S mutations are both located at positions that are distinct from the catalytic sites of POLγ, and structural comparisons revealed that they do not affect the general architecture of the enzyme (Fig. 2a–d). The A467T mutation does not affect the folding of its α-helix (Fig. 2b) and although G848S is relatively close (approximately 5.5 Å) to a backbone phosphate group in the DNA template strand (Fig. 2c), it does not directly interact with the DNA. The serine does form a hydrogen bond with the backbone carbonyl group of V845 (Fig. 2c), but this does not affect the secondary structure relative to the wild-type.

In the presence of PZL-A, we identified a novel density between the POLγA subunit and the proximal POLγB subunit (Fig. 2e and Extended Data Fig. 5a). The size and shape of this density enabled unambiguous placement of PZL-A (Fig. 2f and Extended Data Fig. 5b). PZL-A adopts a C-shaped orientation owing to a stabilizing intramolecular hydrogen bond (2.7–2.8 Å) between the urea and the adjacent pyrazole ring (Fig. 2g). Furthermore, the urea carbonyl group forms a hydrogen bond (approximately 2.7 Å) with the backbone amine of G588 in POLγA (Fig. 2g). Even though the A467T mutation is positioned only around 9 Å from the compound binding site (Extended Data Fig. 5c), the structures of A467T-PZL-A and G848S-PZL-A are almost identical to the apo structures, including wild-type POLγ, with only minor effects on the architecture of the binding pocket or the mutated sites (Fig. 2g–i and Extended Data Figs. 5d,e, 6a–c).

Further structural analysis revealed that the PZL-A-binding pocket is predominantly hydrophobic, with the exception of a small hydrophilic patch near the polar urea carbonyl group and the chromane oxygen of PZL-A (Extended Data Fig. 7a). We identified three residues—L566, H569 and W585—that are in close proximity to PZL-A. The two hydrophobic residues, L566 and W585, support PZL-A binding by flanking the compound on either side, whereas H569 stabilizes the chromane group by contributing to the hydrophilic patch (Extended Data Fig. 7b). Substitution of the three residues with alanine abolished the PZL-A induced thermal shift (Extended Data Fig. 7c), confirming their importance for PZL-A binding. Moreover, PZL-A did not affect the thermal stability of POLγA or POLγB in isolation (Extended Data Fig. 7d). Our data demonstrate that PZL-A binds to a hydrophobic pocket between POLγA and POLγB, a site that is generally unaffected by most disease-causing mutations (https://tools.niehs.nih.gov/polg/).

## PZL-A stimulates POLγ processivity

The location of PZL-A in the ternary complex suggested that it could influence the interplay between POLγA and its associated processivity factor POLγB[13]. Therefore, we monitored DNA synthesis over time using the template in Fig. 1b, but with a radiolabelled primer. Wild-type POLγ produced full-length, double-stranded products of 7.2 kb in less than 15 min, whereas mutant forms of the enzyme (R232H, A467T, W748S and G848S mutants) only produced a smear of shorter products and did not synthesize the full template. Addition of PZL-A had a marked stimulatory effect, leading to DNA synthesis of longer double-stranded DNA (dsDNA) products at levels approaching those seen with wild-type POLγ (Fig. 3a).

To specifically monitor effects on processivity, we conducted a primer extension assay in the presence of heparin. Processivity refers to the average number of nucleotides added by a polymerase in a single binding event, and heparin helps to trap polymerase molecules that dissociate from the template after initiation of DNA synthesis, thereby preventing POLγ from rebinding and continuing to extend the primer (Fig. 3b). All four mutant POLγ variants displayed a distinctly lower processivity than the wild-type control, forming short replication products. Addition of PZL-A had a strong stimulatory effect, restoring processivity to near wild-type levels across all mutant forms of POLγ tested (Fig. 3c).

In agreement with the observed effect on processivity, we noted that PZL-A stabilizes formation of a complex between POLγ and a primed DNA template. In their stalled, elongating conformations, the mutant forms of POLγ—except W748S—displayed lower affinity (increased $K_d$) for the template compared with the wild-type enzyme. The addition of PZL-A had a mild stabilizing effect on all mutants (Extended Data Fig. 8a,b).

To ensure high replication accuracy, DNA polymerases switch between polymerase and exonuclease modes during active DNA synthesis[11,14,15]. To investigate whether PZL-A enhances the stability of the POLγ–primer template complex under these dynamic conditions, we initiated DNA synthesis on radiolabelled primer template, but omitted two nucleotides, causing POLγ to idle at the primer terminus. Excess cold primer template (50-fold) was then added, and the stability of the complex ($k_{off}$) was monitored using an electrophoretic mobility shift assay (EMSA). The mutant POLγ variants exhibited a higher $k_{off}$ than the wild-type, which was reduced upon the addition PZL-A (Extended Data Fig. 8c–e). Together, our processivity assay, $K_d$ measurements and competition experiments demonstrate that PZL-A stabilizes the interaction between mutant POLγ variants and the template during active DNA synthesis.

## PZL-A stimulates the replisome

In living cells, POLγ requires the assistance of the mitochondrial DNA helicase (TWINKLE) and the mitochondrial ssDNA-binding protein (mtSSB). The three factors constitute a minimal replisome that functions on double-stranded mtDNA[16,17]. To investigate the effects of PZL-A on replisome function, we used an approximately 4-kb-long dsDNA template with a preformed replication fork to which the mitochondrial replication machinery could load (Fig 3d). We monitored DNA synthesis by incorporation of radioactive nucleotides over time. As previously reported, the wild-type replisome can perform efficient rolling-circle DNA synthesis on this template, resulting in the formation of DNA products of up to around 20 kb in length[16] (Fig. 3e). All mutant POLγ variants caused a severe defect in DNA synthesis, producing only low levels of shorter replication products in the experimental time window. The addition of PZL-A had a marked stimulatory effect on the

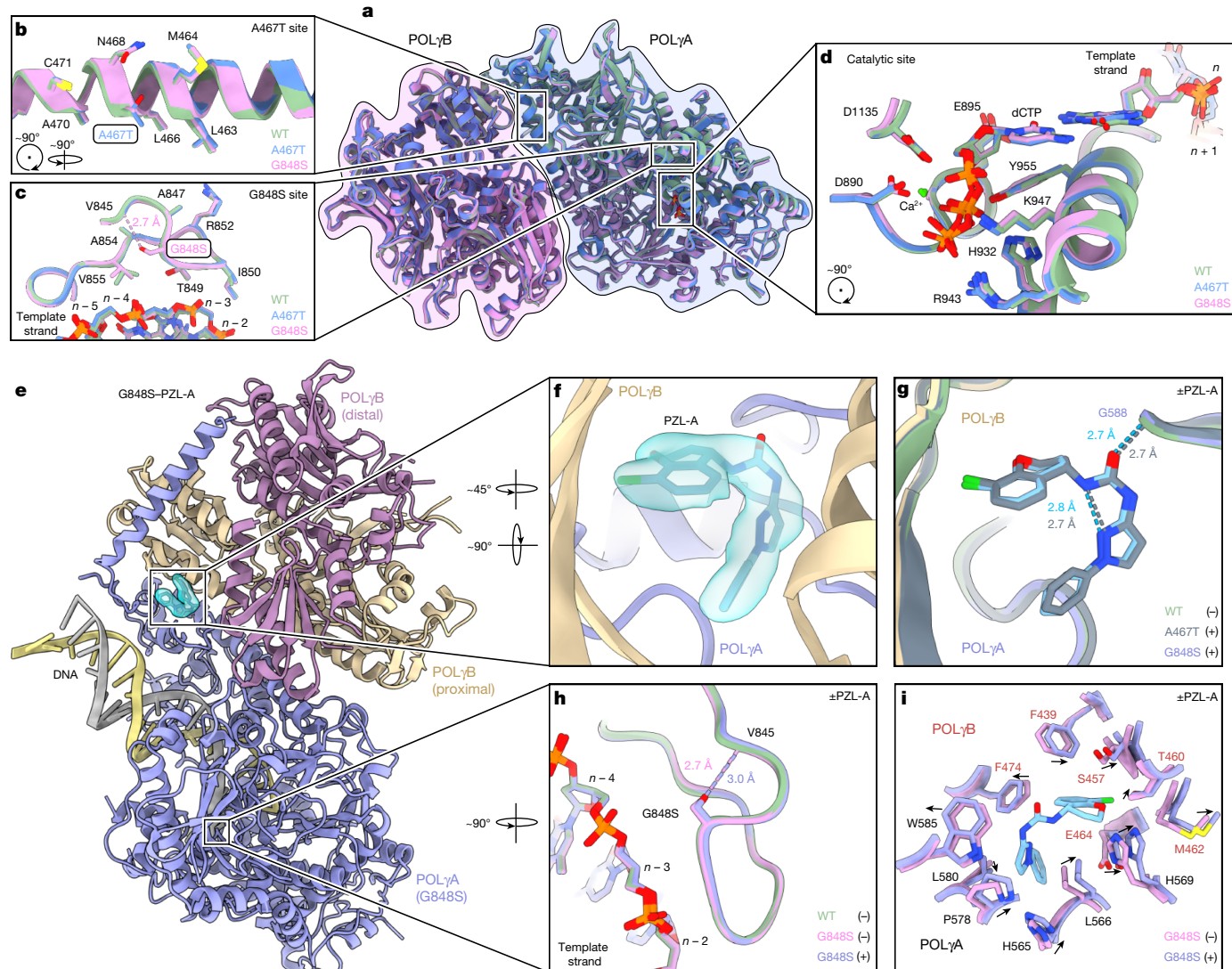

**Fig. 2 | PZL-A binds at the interface between POLγA and POLγB. a**, Overview of overlaid POLγ(A467T) (blue), POLγ(G848S) (pink) and wild-type POLγ (green) heterotrimeric structures. **b**, Close-up of A467, A467T and nearby residues. **c**, Close-up of G848, G848S, DNA and nearby residues. The base pairs in the DNA template strand are numbered according to their proximity to the nucleotide insertion site ($n$). The introduced serine forms a hydrogen bond (dashed line) with the backbone carbonyl of V845. **d**, Close-up of the polymerization site while stalled during elongation. The incoming dCTP base pairs with the template strand at the nucleotide insertion site ($n$). **e**, Cartoon representation of the cryo-EM structure of PZL-A bound to POLγ(G848S). The binding pocket is highlighted with the cryo-EM density for PZL-A (cyan). PZL-A binds POLγ in a pocket at the interface between POLγA and the proximal POLγB subunit.

**f**, Close-up of PZL-A and its cryo-EM density in the binding pocket. **g**, The structures of G848S–PZL-A (colours as in **e**,**f**) and A467T-PZL-A (grey) superimposed on wild-type POLγ (green). PZL-A forms a hydrogen bond with POLγA (G588, 2.7 Å) and an intramolecular hydrogen bond between the urea and pyrazole groups (2.7–2.8 Å). **h**, G848S (pink) and G848S–PZL-A (light purple) structures overlaid on the wild-type POLγ (green) structure at the mutation site. The secondary structure is not affected by the mutation or PZL-A. **i**, Superimposition of the G848S (pink) and G848S–PZL-A (light purple) structures at the binding site of PZL-A. Minor positional changes of nearby residues can be observed upon binding of PZL-A, and the directions of these adjustments are shown with arrows.

DNA synthesis rate, leading to the formation of replication products at levels similar to those observed with wild-type POLγ (Fig. 3e and Extended Data Fig. 8f).

## PZL-A restores mtDNA in cells from patients

To determine whether PZL-A stimulates mtDNA synthesis in cells, we analysed effects of PZL-A on fibroblasts isolated from patients with *POLG* disorders. The cell lines used were compound heterozygous for the mutations that we had previously characterized in vitro−A467T/G848S and W748S/R232H, respectively. In patients, these combinations of mutations cause severe mtDNA depletion and pathological phenotypes, often leading to death in infancy[18]. In the patient cell lines

that we used, the amounts of POLγA were similar to those observed in wild-type controls (Extended Data Fig. 9a). We first verified that PZL-A was well tolerated in patient fibroblasts by assessing cell viability (Extended Data Fig. 9b). We next grew the cells for four days in the presence of ethidium bromide (EtBr), an intercalating compound that selectively depletes mtDNA without affecting nuclear DNA[19]. In this way, we could deplete mtDNA in a controlled manner. After removal of EtBr, the repopulation of mtDNA was monitored over time using quantitative PCR (qPCR) (Fig. 4a). Repopulation was delayed in fibroblasts with *POLG* mutations. Addition of PZL-A did not affect wild-type cells, but strongly stimulated mtDNA repopulation rates in A467T/G848S and W748S/R232H fibroblasts (Fig. 4b−d). These experiments were also repeated in non-dividing, quiescent fibroblasts, in which dNTP levels

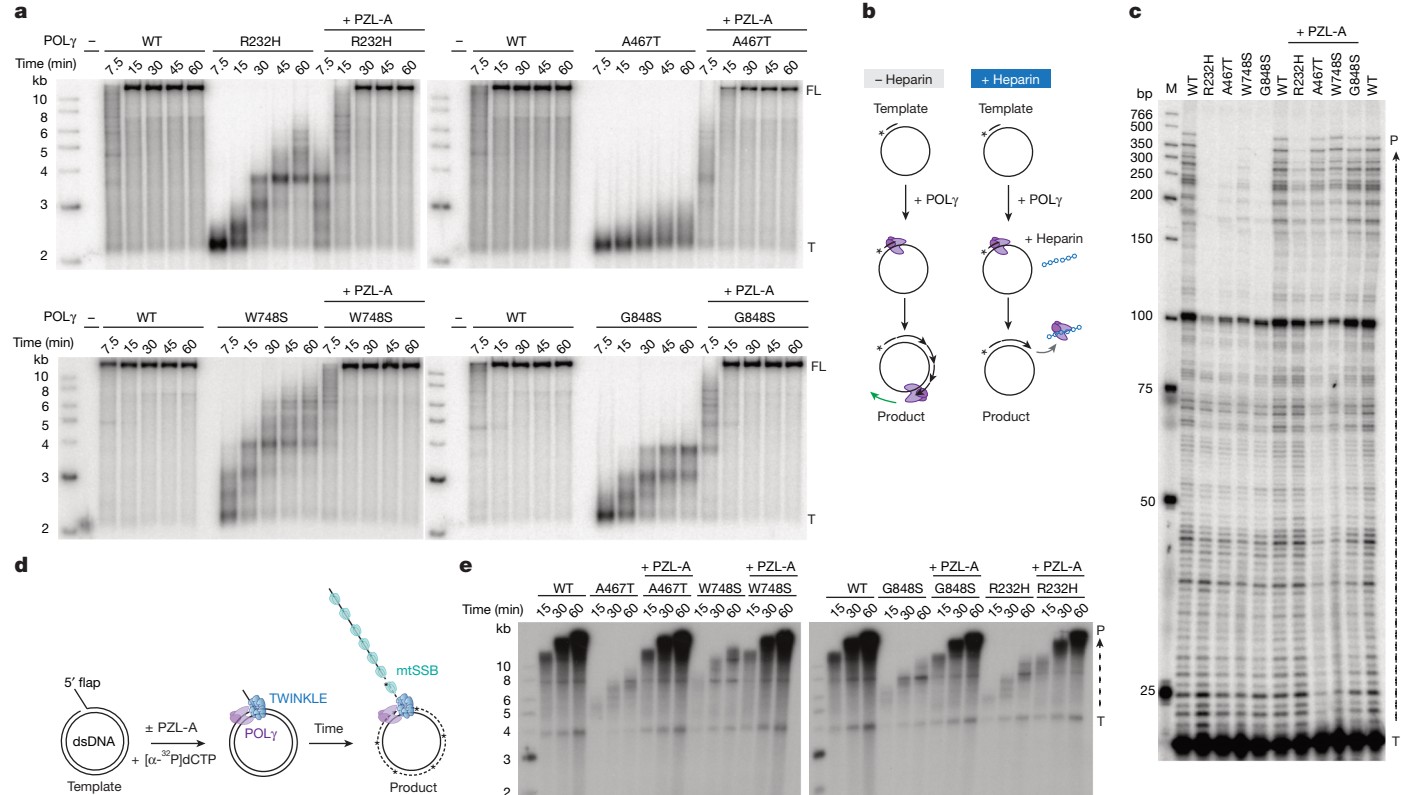

**Fig. 3 | PZL-A increases processivity of mutant POLγ and restores mitochondrial replisome function. a**, Mutant POLγ variants and mtSSB were incubated with a primed, single-stranded DNA (ssDNA) template at 37 °C. Samples were taken at the indicated time points and analysed on a 0.8% native agarose gel. The positions of the full-length (FL) product and the radiolabelled primer template (T) are indicated. Primers may dissociate from the circular ssDNA during electrophoresis, explaining the lower signals in the template control lanes (−). Representative gels are shown (*n* = 3 independent experiments). **b**, Schematic representation of the processivity assay. Heparin is used to trap free POLγ, preventing dissociated enzyme from rebinding to the template to continue DNA synthesis. **c**, Mutant forms of POLγ have impaired processivity compared with wild-type. Processivity is restored in the presence of PZL-A. The products were separated on an 8% urea-PAGE sequencing gel. The positions of products (P) and the radiolabelled template (T) are indicated. Representative gels are shown (*n* = 3 independent experiments). **d**, Schematic representation of the rolling-circle assay used to investigate the effects of PZL-A on replisome function on a dsDNA template. **e**, Time-course experiments as outlined in **d** demonstrate that PZL-A activates mutant POLγ variants during TWINKLE DNA helicase-dependent rolling-circle DNA replication. DNA synthesis was monitored by the incorporation of radiolabelled nucleotides and the resulting products of increasing length were separated on a 0.8% alkaline agarose gel (quantified in Extended Data Fig. 8f). Representative gels are shown (*n* = 3 independent experiments). For source data of gels in **a**,**c**,**e**, see Supplementary Figs. 4–7. Drawings in **b**,**d** by Jennifer Uhler (copyright holder).

are approximately tenfold lower, mimicking the conditions in postmitotic tissues[20] (Extended Data Fig. 9c). Notably, PZL-A also stimulated mtDNA repopulation under these conditions (Extended Data Fig. 9d–f).

To monitor the dose-dependent effects of PZL-A on mtDNA repopulation and genome integrity in dividing cells, we used Southern blotting (Fig. 4e,f and Extended Data Fig. 9g) in combination with qPCR (Extended Data Fig. 9h–j). We observed a clear stimulation of full-length mtDNA repopulation with the addition of as little as 10 nM of the compound to the EtBr-treated cell lines (Fig. 4e,f). Additionally, we noted effects on 7S DNA, a replication intermediate that is lost under conditions when mtDNA synthesis is impaired[21]. EtBr depletion led to a decrease in 7S DNA, but after the addition of PZL-A, the replication intermediate reappeared at levels similar to those seen in wild-type cells (Fig. 4e,f and Extended Data Fig. 9g). To verify that the observed activation was owing to direct effects and not secondary to other cellular events, we isolated mitochondria from A467T/G848S and W748S/R232H fibroblasts and monitored mtDNA synthesis by incorporation of radiolabelled dCTP. As anticipated, de novo mtDNA synthesis was reduced in the mitochondria isolated from patient cells. Addition of PZL-A had a stimulatory effect, demonstrating that the compound could enter mitochondria to directly activate mtDNA synthesis (Fig. 4g).

The stimulatory activity of PZL-A was also evident in downstream effects on the expression of mtDNA in the patient fibroblasts. After EtBr depletion and repopulation for 7 days in the presence of PZL-A, the levels of mtDNA-encoded OXPHOS subunits were increased compared with vehicle-treated controls (Fig. 4h). Consequently, compound treatment also affected OXPHOS activity, increasing basal and maximal respiration as measured by oxygen consumption rates, as well as ATP synthesis (Fig. 4i–k and Extended Data Fig. 9k,l).

Finally, we conducted experiments using neural stem cells carrying the severe G848S mutation in homozygous form. Neural stem cells were selected because they represent a biologically relevant cell type for mitochondrial diseases, given their critical role in neural development and high sensitivity to mitochondrial dysfunction. In these cells, PZL-A increased mtDNA levels and ameliorated OXPHOS activity (Extended Data Fig. 9m–q).

## Discussion

Rare genetic disorders are often caused by diverse mutations within the same gene, disrupting gene function and complicating the development of treatments. Here we show that a single small-molecule activator can restore function across a broad spectrum of *POLG* mutations. Specifically, PZL-A binds to an allosteric site formed at the interface between POLγA and POLγB, which is unaffected by the most common disease-causing mutations[2]. Binding of PZL-A increases the stability

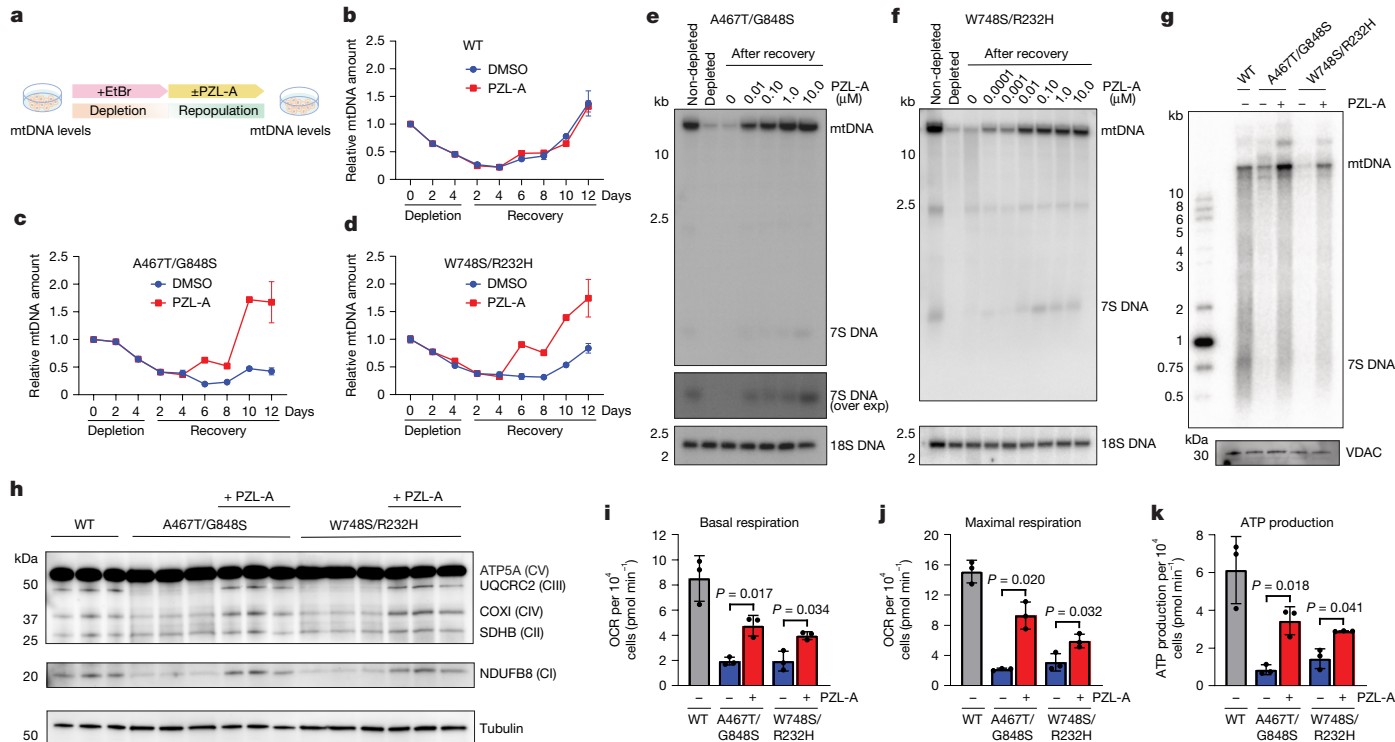

**Fig. 4 | PZL-A stimulates mtDNA synthesis and OXPHOS activity in patient cells. a**, Schematic illustration of the mtDNA depletion and recovery experiment. Drawing by Jennifer Uhler (copyright holder). **b–d**, mtDNA depletion and repopulation in wild-type (**b**), *POLG[A467T/G848S]* (**c**) and *POLG[W748S/R232H]* fibroblasts treated with 50 ng ml[−1] EtBr for 4 days. PZL-A and vehicle were added after EtBr removal. mtDNA was quantified at the indicated time points (mean ± s.d.; *n* = 3 biological replicates). **e**, Dose-dependent recovery of mtDNA in *POLG[A467T/G848S]* mutant cells after 7 days of EtBr treatments (50 ng ml[−1]), followed by recovery for 10 days. PZL-A was added at the indicated concentrations after removal of EtBr. Southern blot analysis after BamHI digestion was used to detect mtDNA and 7S DNA. Nuclear 18S rDNA was used as a loading control. **f**, As in **e**, but with *POLG[W748S/R232H]* mutant cells, 4 days of EtBr treatments and recovery for 5 days. **g**, PZL-A (1 µM) stimulates mtDNA synthesis in intact mitochondria isolated from patient-derived fibroblasts. DNA synthesis was monitored by incorporation of radiolabelled dCTP. VDAC was used as a loading control. **h**, Immunoblot analysis of OXPHOS complex subunits in mutant fibroblasts after 7 days recovery. PZL-A (1 µM) increased NDUFB8 (complex I (CI)), UQCRC2 (complex III (CIII)) and COXI (complex IV (CIV)) levels, whereas SDHB (complex II (CII)) and ATP5A (complex V (CV)) remained unchanged. Tubulin was used as a loading control. **i–k**, Mitochondrial respiration in fibroblast cells treated as in **a**. A Seahorse XFe96 pro extracellular flux analyser was used to measure basal respiration (**i**), maximal respiration (**j**) and ATP production (**k**) after 7 days of recovery. Data are mean ± s.d. (*n* = 3 biological replicates). Unpaired two-tailed Welch's *t*-tests were used to determine *P* values. Blots in **e–h** are representative of *n* = 2 independent experiments. For source data of gels and blots in **e–h**, see Supplementary Fig. 8.

of POLγ bound to template DNA, evidenced by a reduction in $K_d$ and $k_{off}$, and enhances the $k_{cat}$ of the enzyme. The ability of PZL-A to stimulate different mutant forms of POLγ suggests that the pathogenic enzyme variants share certain characteristics. *POLG* is an essential gene, and a common feature of *POLG* patient mutations is that they retain some level of polymerase activity, which is mildly stimulated by higher dNTP concentrations (Extended Data Fig. 1a). We propose that elevated dNTP levels in dividing cells, which are 10–20 times higher than in non-dividing cells[22], help to ameliorate the effects of many *POLG* mutations, enabling them to be tolerated during embryogenesis. The mutant phenotypes become apparent when cells stop dividing, dNTP levels drop and postmitotic tissues such as muscle and neurons are formed. In support of this notion, previous studies have shown that the addition of large amounts of dNTPs can mitigate mtDNA depletion in fibroblasts with various *POLG* mutations[20]. Rather than completely inactivating the DNA polymerase activity, the common *POLG* mutations analysed here impair both catalytic efficiency and processivity. This explains why PZL-A, which enhances POLγ–DNA binding stability and increases the catalytic turnover of the enzyme, can broadly restore POLγ function across different pathogenic mutations.

In future research, we plan to expand our analysis to include additional POLγ mutations to further delineate the boundaries of our approach. We will also investigate whether the compounds can prevent formation of deleted mtDNA molecules, typically seen in older patients with milder forms of *POLG* disorder[2]. Although the present study focuses on disease-causing mutations, our initial screen also revealed that our compound can stimulate wild-type POLγ in vitro. This activity could potentially offer a solution to mtDNA replisome deficiencies arising from non-*POLG* mutations, such as those affecting the TWINKLE DNA helicase or genes involved in mitochondrial nucleotide metabolism[23]. Additionally, it would be informative to study the effects of PZL-A on the gradual depletion of mtDNA and the accumulation of deleted mtDNA molecules, which are associated with more general conditions, including neurodegenerative disorders and ageing[4].

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

## Methods

### Protein expression and purification

POLγA (P54098), wild-type and mutant variants (R232H, A467T, W748S+E1143G, and G848S), POLγB (AAD56640.1) and the TWINKLE DNA helicase (Q96RR1) were cloned and expressed as 6×His-tagged fusion proteins in *Spodoptera frugiperda* cells, following previously published protocols[24]. The *SSBP1* gene encoding the human mtSSB (Q04837) was cloned into the pET-17b vector in frame with a C-terminal 6× His-tag. The mtSSB protein was subsequently purified as previously reported[25].

### DNA synthesis on ssDNA template

A 20-mer oligonucleotide [32]P-labelled at the 5′ end (5′-GTAAAACGACG GCCAGTGCC-3′) was hybridized to M13mp18 ssDNA (New England Biolabs). Reactions were performed in 20 µl volumes, each containing 0.5 nM of template DNA, 25 mM Tris-HCl pH 8.0, 1 mM Tris (2-carboxyethyl)phosphine (TCEP), 10 mM MgCl$_2$, 0.1 mg ml$^{-1}$ bovine serum albumin (BSA), 35 mM NaCl, 10 µM of all four dNTPs (unless otherwise indicated in the figure), 1% DMSO, 200 nM mtSSB (calculated as a tetramer), 2.5 nM of the specified POLγA variants, 7.5 nM (5 nM was used in the time-course experiment) POLγB (concentration calculated as a dimer), and 1 µM PZL-A when indicated. The reactions were incubated at 37 °C for the specified duration and terminated by the addition of 4 µl stop buffer (90 mM EDTA pH 8.0, 6% SDS, 30% glycerol, and 0.25% bromophenol blue). Products were separated on a 0.8% agarose gel with 0.5 µg ml$^{-1}$ EtBr at 40 V in 1× TBE buffer for 18 h and visualized by autoradiography.

The 20-mer oligonucleotide hybridized to M13mp18 ssDNA template was also used for SYBR Green assays but without radioactive 5′-labelling of the primer. Reactions were performed at a final volume of 10 µl. Each reaction contained 0.5 nM of template DNA, 25 mM Tris-HCl pH 8.0, 1 mM TCEP, 25 mM NaCl, 10 mM MgCl$_2$, 0.1 mg ml$^{-1}$ BSA, 100 µM of all 4 dNTPs, 0.02% Triton X-100, 200 nM mtSSB (calculated as a tetramer), 0.5 nM of the specified POLγA variants, and 0.65 nM POLγB (calculated as a dimer). Duplicates of reaction mixtures were distributed into microplates (384-well) with wells containing compounds prepared from 10 mM compound stocks in 100% DMSO (final compound concentrations indicated in the figure). Equal amounts of DMSO (1% in final concentration) were added to positive and negative control reactions. The reactions were incubated at 37 °C for 2 h in a VWR INCU-Line incubator and terminated by the addition of 10 µl of 50 mM EDTA pH 8.0, 0.02% Triton X-100, and SYBR Green I (1:5,000) followed by incubation for 20 min at room temperature. The fluorescence signal was analysed using a BMG PHERAstar microtitre plate reader (between 485–520 nm) and BMG PHERAstar microtitre plate reader control software. The signal was normalized relative to wild-type (1% DMSO) and the data were plotted in Prism 10 (Graphpad Software), with errors shown as s.d., and fitted using the '[agonist] versus response (three parameters)' model.

### Identification of PZL-A

To identify activators that enhance POLγ activity, we screened a diverse set of about 270,000 small-molecule compounds. We adopted a homogeneous fluorescence-based method[26], in which a fluorescent reporter strand is annealed to a longer, quencher-labelled template strand. DNA synthesis is initiated from a primer annealed to the same template strand and displacement of the short reporter strand is measured as an increase of fluorescence. In the initial screen we used wild-type POLγ and the threshold for hit selection was set stringently to 30% increased activity, relative to the control.

Changes in dose–response activity for wild-type POLγ and mutant derivatives during further compound optimization were assessed using the SYBR Green fluorescence assay described above. A variety of follow-up assays described in this Article were used to assess target engagement, selectivity, and cellular effects.

### Steady-state kinetics

To determine apparent $K_{m\,app,\,dNTP}$ and $k_{cat}$, we used a 20-mer oligonucleotide hybridized to an M13mp18 ssDNA template (described above) and performed time-course experiments at 8 different concentrations of dNTPs (0.08, 0.16, 0.32, 0.625, 1.25, 2.5, 5 and 10 µM) and monitored incorporation of dNTPs. Reactions were performed in 160 µl volumes containing 0.5 nM of DNA template, 25 mM Tris-HCl pH 8.0, 1 mM TCEP, 35 mM NaCl, 10 mM MgCl$_2$, 0.1 mg ml$^{-1}$ BSA, 1.3 nM POLγA, 3.75 nM POLγB (calculated as a dimer), 200 nM mtSSB (calculated as a tetramer) and 1 µM PZL-A or 1% DMSO, and the indicated concentrations of dNTPs. To follow nucleotide incorporation, dNTPs were spiked with a small amount of [α-32P]dCTP (40 µCi (≈13 pmol) was added to 100 µl of a 200 µM dNTP solution). This dNTP solution was then diluted accordingly, both for the reactions and the calibration curve. Reactions were incubated at 37 °C and 20 µl samples were taken at 0, 2.5, 5, 7.5, 10 and 15 min and stopped by the addition of 5 µl of 0.5 M EDTA pH 8.0. The products were analysed by placing 5 µl onto a Hybond N$^+$ positively charged membrane. The membrane was air-dried for 15 min and then washed 3 × 5 min in 2× saline-sodium citrate (SSC) buffer followed by a 2 min wash in 95% ethanol to remove non-incorporated nucleotides, air-dried again and visualized by autoradiography and quantified using the Fujifilm Multi Gauge V3.1 software. A standard curve was generated by adding 5 µl of each indicated amount of dNTP onto a membrane, which was dried without washing representing the total amount of dNTPs.

To determine the apparent $K_d$ for interactions between PZL-A and POLγ ($K_{d\_app,\,PZL-A}$), we performed time-course experiments (0, 2.5, 5, 7.5, 10 and 15 min) at different concentrations of PZL-A (1.37, 4.1, 12.3, 37, 111, 333 and 1,000 nM) with a fixed concentration of 20 µM dNTP (including 0.4 µCi (≈133 fmol) [α-32P]dCTP) (Table 1). The experiments were performed as described for the steady-state kinetics experiments.

Initial rates were determined for the different dNTP and PZL-A concentrations used. For the dNTP titration experiments, the rates were normalized by dividing by the concentration of active enzyme–DNA complex (see 'Determination of active POLγ–DNA complex'). Velocity was plotted (using Prism 10) against [dNTP] or [PZL-A], and non-linear regressions were performed using either the 'Michaelis–Menten' or 'one site–specific binding' models to obtain values for $K_{m\,app,\,dNTP}$ and $k_{cat}$ or $K_{d\_app,\,PZL-A}$. All reactions were performed at least three times.

### Determination of active POLγ–DNA complex

The concentration of active enzyme–DNA complex was determined by performing EMSA as described in 'The DNA-binding activity' but using the same conditions as in the kinetics assays. This was accomplished by using 0.5 nM template DNA in presence of 1.3 nM POLγA and increasing amounts of POLγB (0, 5, 10, 20, 40, 60, 100 and 200 fmol POLγB in 15 µl reactions). While POLγA alone binds the DNA and generates a shift, the addition of POLγB generates a slower migrating shift which represents the active complex. As the amount of POLγB increases, the amount of active protein also increases until it reaches a plateau (saturated POLγA). At the POLγB:POLγA ratio 3:1 (POLγB calculated as dimer) the system reached the conditions used in the kinetics assays. By quantification of bound and unbound DNA we could determine the concentration of active protein–DNA complex (Extended Data Fig. 1b). These values were used to normalize the rates in the steady-state kinetics assay (dNTP titration) and to calculate the $V_{max}$ as presented in Table 1.

### Exonuclease activity

To measure the 3′-to-5′ exonuclease activity of POLγA, a 20-mer oligonucleotide (5′-GCGGTCGAGTCCGGCGGCGC-3′) was [32]P-labelled at the 5′ end and annealed to a 36-mer oligonucleotide (5′-GACTACGTCTATC CGGACGCCGCCGGACTCGACCGC-3′), resulting in a 19-bp dsDNA region with a single-nucleotide mismatch at the 3′ end. Reaction mixtures (10 µl) contained 25 mM Tris-HCl pH 8.0, 1 mM DTT, 10 mM MgCl$_2$,

0.1 mg ml$^{-1}$ BSA, 8.7% glycerol, 1 nM template, 2 nM POLγA, 4 nM POLγB (calculated as a dimer), 1% DMSO, and when indicated, 1 μM PZL-A. Reactions were incubated at 37 °C for the indicated duration and terminated by the addition of 10 μl stop buffer (98% formamide, 10 mM EDTA pH 8.0, 0.025% bromophenol blue, and 0.025% xylene cyanol). The products were separated by electrophoresis in 7 M urea/10% polyacrylamide gels for 2 h at 1500 V and visualized by autoradiography. The time required for the exonuclease activity to degrade the mismatch nucleotide (C), and the subsequent nucleotide (G), was quantified using the Fujifilm Multi Gauge V3.1 software. s.d. values were calculated from triplicate experiments to represent variability.

### Differential scanning fluorimetry

The fluorescent dye SYPRO Orange was used to monitor temperature-induced unfolding of POLγ as previously described[27]. In brief, the experiment was performed in 384-well PCR plates and individual reactions contained wild-type or mutant proteins (final concentration 0.5 μM), 5× SYPRO Orange, 20 mM Tris-HCl pH 8.0, 100 mM NaCl and 1 mM DTT, in the absence or presence of 10 μM PZL-A. Differential scanning fluorimetry was performed in a CFX Opus 384 Real-Time PCR System using the CFX Maestro real-time software (Bio-Rad). Scans were recorded using the HEX emission filter (560–580 nm) between 4 and 95 °C in 0.5 °C increments with a 5 s equilibration time. The melting temperature ($T_m$) was determined from the first derivative of a plot of fluorescence intensity versus temperature. $\Delta T_m$ was determined by subtracting $T_m$ (without PZL-A) from $T_m$ (with PZL-A). The s.d. of $\Delta T_m$ was calculated from three independent measurements.

### Cryo-electron microscopy sample preparation and data acquisition

Primer template DNA was formed by annealing a 25-nt primer (5′-GCATGCGGTCGAGTCTAGAGGAGCC-3′) to a 40-nt template strand (5′-TTTTTTTTTTTATCCGGGCTCCTCTAGACTCGACCGCATGC-3′). To prepare cryo-EM samples, POLγA (A467T, G848S, or wild-type) was mixed at a 1:2 molar ratio with POLγB and dialysed into a buffer containing 20 mM HEPES-NaOH pH 7.5, 140 mM KCl, 10 mM CaCl$_2$ and 1 mM TCEP. Following dialysis, PZL-A was added to a final concentration of 20 μM and incubated for 10 min at room temperature (this step was omitted for the apo samples). Primer template DNA was added at a POLγ:DNA molar ratio of 1:1.2, and dCTP was added to a final concentration of 0.2 mM. The samples were incubated on ice for 10 min before being applied to the grids. Grids were prepared using a Vitrobot Mark IV (Thermo Fisher Scientific) by adding 3.5 μl of approximately 2 μM protein sample to glow-discharged QuantiFoil 2/1 grids (Quantifoil Micro Tools GmbH) at 4 °C and 100% humidity. Grids were blotted for 2 s and plunge-frozen in liquid ethane.

Cryo-EM data were collected on two Titan Krios G2 microscopes (Thermo Fisher Scientific) at the SciLifeLab cryo-EM facility in Stockholm, both operated at 300 kV and equipped with a K3 direct electron detector (Gatan) and a BioQuantum energy filter (Gatan) set to a slit width of 20 eV. Five cryo-EM datasets (G848S-PZL-A, G848S, A467T-PZL-A, A467T, and wild-type) were collected in super-resolution mode at a nominal magnification of 105,000×, corresponding to a calibrated pixel size of 0.825 Å or 0.828 Å. The total dose was 40–42 e$^-$ Å$^{-2}$ per 40 frames, and the movies were acquired at a defocus range of −0.8 μm to −2.2 μm.

### Cryo-electron microscopy data processing

The processing workflows for G848S-PZL-A, G848S, A467T-PZL-A, A467T, and wild-type POLγ are shown in Extended Data Figs. 2–4. For the five datasets, the acquired movie stacks were imported into cryoSPARC (v4.3.1) for image processing[28]. Motion correction of the data was performed with Patch Motion Correction, and the contrast transfer function (CTF) was estimated with Patch CTF Estimation. Micrographs with poor CTF fit (worse than 4 Å) were removed.

The Automatic Blob Picker was used to pick particles, which were extracted with 4× binning (3.3 Å per pixel). Multiple rounds of two-dimensional (2D) classifications were performed to filter out junk particles. The remaining particles were used to perform a three-class ab initio reconstruction followed by heterogeneous refinement. At this stage, for all five datasets, there was at least one class with clear POLγ features. These classes were kept and filtered further with several rounds of 2D classifications and heterogeneous refinements with one junk volume to remove poor particles. The particles and volume from the POLγ class for each of G848S-PZL-A, A467T-PZL-A, and A467T were refined with non-uniform refinements and local/global CTF refinements, resulting in final 3D reconstructions with a gold-standard Fourier shell correlation (GSFSC) resolution of 2.63 Å, 2.67 Å and 2.69 Å, respectively. During the processing of G848S and wild-type POLγ, two POLγ classes emerged after heterogeneous refinement. However, in both cases, one of these classes was anisotropic due to accumulation of side views and not used for the final reconstructions. The other class, for both G848S and wild-type, was refined with non-uniform refinements and local/global CTF refinements, resulting in final 3D reconstructions with a GSFSC resolution of 2.39 Å and 2.65 Å, respectively. The GSFSC = 0.143 criterion was used for determining the resolution of all reconstructions. All final maps were sharpened using DeepEMhancer and Phenix Autosharpen[29,30].

### Model building, refinement and analysis

To build the wild-type and mutant POLγ structures, the POLγ ternary complex (PDB ID: 4ZTZ) was docked into the cryo-EM maps by rigid body fitting in UCSF ChimeraX (v.1.4) and manually fitted in real space in Coot (v.0.9.8.1) and ISOLDE (v.1.4)[31–33]. In the mutant models (A467T and G848S), the amino acids were mutated in Coot. In the cryo-EM maps generated from samples with PZL-A added, an unaccounted density was identified between the POLγA and the proximal POLγB unit, where PZL-A could be fitted. Coordinates and restraints for PZL-A were created using the eLBOW module in PHENIX (v.1.20) and positioned in the density[34,35]. After initial fitting, the models were improved iteratively with real-space refinement in PHENIX and manual adjustments in Coot. The refined models were validated with MolProbity (Extended Data Table 1)[36]. Local resolution was estimated for all five maps with cryoSPARC (Fourier shell correlation (FSC) threshold = 0.5) (Extended Data Figs. 2–4). Figures were prepared using UCSF ChimeraX.

### The DNA-binding activity

DNA binding of POLγ holoenzyme (POLγA and POLγB in complex) to a primer template was assayed using a 36-mer oligonucleotide (5′-TTTTTTTTTTTATCCGGGCTCCTCTAGACTCGACCGC-3′) annealed to a 5′-$^{32}$P-labelled 21-mer complementary oligonucleotide (5′-GCG GTCGAGTCTAGAGGAGCC-3′). This produces a primed template with a 15-base single-stranded 5′ tail. Reactions were carried out in 15 μl volumes containing 0.5 nM DNA template, 25 mM Tris-HCl pH 8.0, approximately 30 mM NaCl, 1 mM TCEP, 0.1 mg ml$^{-1}$ BSA, 10% glycerol, 10 μM ddCTP and either 1% DMSO or 1 μM PZL-A. POLγB (6.6 nM, calculated as a dimer) was included in the mixture and POLγA (0, 0.011, 0.021, 0.042, 0.083, 0.17, 0.33, 0.67 and 1.33 nM) was added as indicated in the figures. Reactions were incubated on ice for 10 min followed by 10 min at room temperature before separation on a 6% native PAGE gel in 0.5× TBE for 40 min at 180 V. Bands were visualized by autoradiography.

For $K_d$ analysis, band intensities representing unbound and bound DNA were quantified using Multi Gauge V3.0 software (Fujifilm Life Sciences). The fraction of bound DNA was calculated from the background-subtracted signal intensities using the expression: bound/(bound + unbound). The fraction of DNA bound in each reaction was plotted against the concentration of POLγ. Data were fitted using the quadratic equation (fraction DNA bound = (([POLγ$_{tot}$] + [DNA$_{tot}$] + $K_d$) − sqrt(([POLγ$_{tot}$] + [DNA$_{tot}$] + $K_d$)$^2$ − 4[POLγ$_{tot}$] × [DNA$_{tot}$]))/2[DNA$_{tot}$]) in Prism 10 (Graphpad Software) to

obtain values for $K_d$. Each $K_d$ value is presented as the average of three independent reactions.

For the off-rate measurements, the same primer template used in the cryo-EM studies was used but the 25-nucleotide primer was $^{32}$P-labelled in the 5′ end. The reactions contained 5 nM of indicated POLγA variants, 7.5 nM POLγB, 0.6 nM $^{32}$P-labelled primed template, 25 mM Tris-HCl pH 7.8, 10 mM MgCl$_2$, 0.1 mg ml$^{-1}$ BSA, 10 mM DTT, 8.5% glycerol, 100 μM dGTP, and 100 μM dCTP or ddCTP in a final volume of 15 μl. When indicated, 1 μM PZL-A was added. The mixture was pre-incubated at room temperature for 10 min, followed by the addition of a 50× excess cold (non-labelled) primer template, and incubated at 37 °C for indicated time. The samples were separated on a 4% native polyacrylamide gel (0.5× TBE) for 20 min at 180 V and visualized by autoradiography. Band intensities representing unbound and bound DNA were quantified using Multi Gauge V3.0 software (Fujifilm Life Sciences). The remaining fraction of bound DNA was determined from the background-subtracted signal intensities, using the expression: bound/(bound + unbound), and normalized against the value at $t = 0$. DNA bound (in %) in each reaction was plotted versus time (min). Data were fit using the "dissociation–one phase exponential decay" model in Prism 10 (Graphpad Software) to obtain the dissociation rate constant, $k_{off}$, with errors shown as s.e.m.

## Processivity measurements

The same template as in DNA synthesis on ssDNA template described above was used in a reaction mixture A (10 μl) containing 50 mM Tris-HCl pH 8.0, 2 mM TCEP, 0.2 mg ml$^{-1}$ BSA, 50 mM NaCl, 2% DMSO, 5 nM template, 2.5 nM of the specified POLγA variants, 5 nM POLγB (calculated as a dimer), and, when indicated, 2 μM PZL-A. The A mixtures were incubated for 10 min on ice prior to adding mixture B (10 μl) containing 20 mM MgCl$_2$, 20 μM of all four dNTPs, and 600 ng ml$^{-1}$ heparin, and then immediately incubated at 37 °C for 5 min. Reactions were stopped by adding 20 μl stop buffer (95% formamide, 20 mM EDTA pH 8.0, and 0.1% bromophenol blue). Samples were heated at 95 °C for 3 min before loaded on a pre-run (1 h, 1,500 V) 8% denaturing PAGE-UREA (6 M Urea, 1× TBE) sequencing gel and run for 75 min at 1,500 V in 1× TBE buffer. Replication products were visualized by autoradiography.

## Rolling-circle replication

A dsDNA template with a preformed replication fork was generated by annealing a 70-mer oligonucleotide (5′-T$_{42}$ATCTCAGCGATCTGTC TATTTCGTTCAT-3′) to single-stranded pBluescript SK(+) O$_L$, followed by one cycle of polymerization with KOD polymerase, as previously described[24]. The template (0.4 nM) was added to a reaction mixture (final volume 25 μl) containing 25 mM HEPES-NaOH, pH 7.6, 10 mM DTT, 10 mM MgCl$_2$, BSA (0.1 mg ml$^{-1}$), 1 mM ATP, 10 μM of all four dNTPs, 2 μCi of [α-$^{32}$P]dCTP, 1% DMSO, 4 nM TWINKLE (calculated as a hexamer), 160 nM mtSSB (calculated as a tetramer), 6 nM POLγA (wild-type or mutant variants), and 9 nM POLγB (calculated as a dimer). When indicated, 1 μM PZL-A was included in the reactions. After incubation at 37 °C for times indicated, reactions were terminated by adding 8 μl of alkaline loading buffer (18% (w/v) Ficoll, 300 mM NaOH, 60 mM EDTA pH 8.0, 0.25% (w/v) bromophenol blue, and 0.25% (w/v) xylene cyanol FF) and separated on a 0.8% denaturing agarose gel at 40 V for 20 h. The experiments were performed in triplicate. Reaction products were visualized by autoradiography and quantified using Fujifilm Multi Gauge V3.1 software. Standard deviations were calculated to represent variability across the triplicate experiments.

## Cell lines

Primary skin fibroblasts from two patients with mutations in *POLG* (NP_001119603.1) were obtained from the Swedish Biobank. Patient 1 exhibited compound heterozygosity for mutations p.[Ala467Thr] and [Gly848Ser], and patient 2 harboured compound heterozygous mutations p.[Trp748Ser+Glu1143Gly] and p.[Arg232His]. Primary skin fibroblasts from donors with wild-type POLγ were used as controls. Genotypes were confirmed via PCR amplification followed by Sanger sequencing. Fibroblasts were cultured in Dulbecco's Modified Eagle Medium (DMEM; 4.5 g l$^{-1}$ glucose, 4 mM glutamine, 110 mg l$^{-1}$ sodium pyruvate) supplemented with 10% fetal bovine serum at 37 °C in a 5% CO$_2$ humidified cell incubator. All cell lines were authenticated by STR profiling, with no common misidentified lines detected. They were routinely tested for mycoplasma contamination using PCR, and all results were confirmed negative.

A human induced pluripotent stem (hiPS) cell line (PGP1-SV1) was reprogrammed from PGP1 fibroblasts and used as a control for all sequential stem cell-related experiments. The PGP1 hiPS cell line was derived from fibroblasts obtained from G. Church as part of the Personal Genome Project (PGP), an initiative dedicated to open-access genomic research. The fibroblast samples were collected with informed consent, explicitly permitting the public posting and unrestricted commercial use of Personally Identifying Genetic Information (PIGI). The donors signed an informed consent form that allows for open-access distribution of genetic and cellular data, including whole-genome SNP arrays, whole-exome sequences, and whole-genome sequences. The PGP1 cells have no commercial restrictions under the Open Material Transfer Agreement and are freely available for research use. The PGP1 cell line has been previously characterized and is described in the patent: Methods for increasing efficiency of nuclease-mediated gene editing in stem cells[37].

The PGP1 fibroblasts were reprogrammed into hiPS cells using a non-integrating Sendai viral approach with the CytoTune-iPS 2.0 Sendai Reprogramming Kit (Thermo Fisher Scientific). This method allows for efficient reprogramming without genomic integration, ensuring the maintenance of genomic stability.

To model *POLG*-related mitochondrial disease, a homozygous *POLG* mutation p.[Gly848Ser] was introduced into the PGP1 hiPS cell line using CRISPR–Cas9 gene-editing technology (Synthego). Mutations were confirmed by PCR amplification followed by Sanger sequencing.

## Cell viability assay

Relative cell viability of primary skin fibroblast cell lines derived from *POLG* patients was assessed using the MTT Cell Proliferation Kit I (Roche Diagnostics, 11465007001) following the manufacturer's protocol. Cells were treated with compound concentrations ranging from 0 to 10 μM (0, 0.003, 0.01, 0.03, 0.1, 0.3, 1, 3 and 10 μM) for 120 h, with treatment starting 24 h after cell seeding. Vehicle (0.1% DMSO) served as a control.

## mtDNA depletion and recovery in proliferating cells

mtDNA depletion was induced by adding 50 ng ml$^{-1}$ EtBr to the culture medium for up to 7 days followed by a return to EtBr-free medium. Cell pellets were collected at time points indicated in figure legends for subsequent DNA extraction during EtBr treatment. After removal of EtBr, cells were maintained at 70–90% confluence to ensure exponential growth and split every 3–4 days with fresh medium containing PZL-A or vehicle (0.1% DMSO). The treatment duration extended up to 12 days, as indicated in the figure. PZL-A was added at concentrations of 3 μM to POLγ(A467T/G848S) or wild-type POLγ cells and 1 μM to POLγ(W748S/R232H) mutant cells, and vehicle served as a control. Cell pellets were collected at specified time points for subsequent DNA extraction. EtBr-untreated cells served as non-depleted controls in parallel experiments. No differences in cell doubling times were observed during EtBr treatment or the repopulation phase of the experiment. All cell culture studies were conducted in triplicate unless otherwise stated in the figure legends.

## mtDNA depletion and recovery in quiescent cells

Fibroblast cells were seeded in 12-well plates, and mtDNA depletion was induced by adding 50 ng ml$^{-1}$ EtBr to the culture medium for

7 days, followed by a return to EtBr-free medium. After 7 days of EtBr treatment, the cells reached confluence. EtBr was then withdrawn, and the FBS concentration was reduced to 0.1% to induce quiescence. Three days later, the cells were treated with either vehicle (0.1% DMSO) or 1 μM PZL-A for 7 or 14 days. During treatment, the medium containing PZL-A or vehicle was refreshed every 3–4 days. DNA was extracted at various time points, as indicated in the figure legends, and mtDNA levels were quantified by qPCR. EtBr-untreated cells served as non-depleted controls in parallel experiments. Experiments were performed in three independent biological replicates.

### Neuronal stem cell culture and treatments
G848S/G848S mutant and PGP1 hiPS cells were cultured and expanded in mTeSR Plus complete medium (Stemcell Technologies, 100-0276) on iMatrix511 coated plates (ReproCell, NP892-011) and passaged using StemPro Accutase (Thermo Fisher Scientific, A1110501). hiPS cells were differentiated into neural stem cells (NSCs) under feeder-free conditions in 3% $CO_2$ (Creative Biolabs).

G848S/G848S mutant and PGP1 NSCs were cultured on Matrigel-coated dishes in StemDiff Neural Progenitor Medium (Stemcell Technologies, 05833) supplemented with 5 μM Y27632 (Stemcell Technologies, 72304). Cells were maintained in a 37 °C humidified incubator with 5% $CO_2$.

NSCs were treated with 0.2 μM or 1 μM PZL-A, or vehicle (0.01% DMSO), for 10 days. During the treatment period, cells were passaged every 3–4 days using Accutase and cultured in fresh medium containing PZL-A or vehicle. After 10 days of treatment, cells were detached with Accutase and replated at 30,000 cells per well onto Matrigel-coated Seahorse XFe96 Pro plates. The Agilent Mito Stress Test was performed as described in the Metabolic Flux Assay section, and cell numbers per well were determined using the Invitrogen CyQuant assay. Parallel cell pellets collected after 10 days of treatment were used for DNA extraction and mtDNA quantification. All experiments were performed in three independent biological replicates.

### mtDNA quantification
Cell pellets were lysed in lysis buffer (100 mM NaCl, 25 mM EDTA pH 8.0, 10 mM Tris-HCl, pH 8.0, 0.5% SDS) and digested with Proteinase K (100 ng ml⁻¹) at 55 °C for 1 h. Genomic DNA was extracted using Zymo Genomic DNA Clean and Concentrator purification kit according to the manufacturer's instructions (Zymo Research, D4010). Prior to elution, DNA was treated with RNase A (100 ng ml⁻¹). The mtDNA quantification was measured using real-time qPCR with iTaq Universal SYBR Green Supermix (Bio-Rad, 1725124). Specific primers targeting the mitochondrial gene MT-CYB (5′-ACAATTCTCCGATCCGTCCC-3′ and 5′-GTGATTGGCTTAGTGGGCGA-3′) were used for mtDNA quantification, and primers for the nuclear 18S rRNA (5′-AGAAACGGCTAC CACATCCA-3′ and 5′-CCCTCCAATGGATCCTCGTT-3′) or *B2M* gene (5′-TGCTGTGTCTCCATGTTTGATGTATCT-3′ and 5′-TCTCTGCTCCCCAC CTCTAAGT-3′) served as a loading control.

Southern blotting was also performed to evaluate mtDNA and 7S DNA levels following mtDNA depletion and recovery in the presence of PZL-A during the proliferation phase. Genomic DNA (2 μg) was digested at 37 °C overnight using 20 U of the BamHI-HF restriction enzyme. Digested DNA was separated by electrophoresis on a 1% agarose gel, followed by depurination in 0.25 M HCl for 20 min. The DNA was denatured in 0.5 M NaOH and 1.5 M NaCl, neutralized in 0.5 M Tris-HCl, pH 7.4, and 1.5 M NaCl, and transferred overnight onto a Hybond N+ nitrocellulose membrane. Cross-linking was performed using 254-nm ultraviolet light at 200 mJ cm⁻². The membrane was sequentially hybridized with probes detecting mtDNA and nuclear 18S DNA. Radiolabelled dsDNA probes were prepared by random primer labelling (Agilent, 300385) of gel-extracted human mtDNA (5′-CTCACCCACTAGGATACCAAC-3′ and 5′-GATACTGCGACATAGGGTGC-3′) and 18S rDNA (5′-AGAAACG GCTACCACATCCA-3′ and 5′-CCCTCCAATGGATCCTCGTT-3′) PCR

products. After hybridization, radiolabelled signals were detected using a FLA-7000 phosphorimager (version 1.12) and analysed with Multi Gauge software (version 3.0, Fujifilm).

### In organello replication
Freshly isolated mitochondria (1 mg) were resuspended in 1 ml of incubation buffer (25 mM sucrose, 75 mM sorbitol, 100 mM KCl, 10 mM $K_2HPO_4$, 0.05 mM EDTA pH 8.0, 5 mM $MgCl_2$, 1 mM ADP, 10 mM potassium glutamate, 2.5 mM potassium malate, 10 mM Tris-HCl, pH 7.4) supplemented with 1 mg ml⁻¹ fatty acid-free BSA. The mixture was supplemented with 50 μM each of dTTP, dATP, and dGTP along with 20 μCi of [α-³²P]dCTP (3,000 Ci mmol⁻¹) and incubated at 37 °C for 2 h on a rotating wheel. Following incubation, mitochondria were pelleted at 7,600g for 4 min and washed twice with washing buffer (10% glycerol, 10 mM Tris-HCl, pH 6.8, 0.15 mM $MgCl_2$). mtDNA was extracted using phenol followed by ethanol precipitation.

Resuspended mtDNA samples were heated to 95 °C for 2 min to release 7S DNA from mtDNA and then separated on a 1% agarose gel. The dried agarose gel was exposed to a phosphorimager screen for visualization. For loading controls, protein extracts from equal aliquots of mitochondria were prepared and subjected to VDAC immunoblotting.

### Immunoblotting
Cell pellets were incubated in a cell lysis buffer (50 mM Tris-HCl, pH 7.4, 150 mM NaCl, 1 mM EDTA pH 8.0.1% Triton X-100, supplemented with 1× protease inhibitors: 2 mM benzamidine, 2 μM pepstatin A, 1 mM phenyl-methylsulfonylfluoride, 0.5 μM leupeptin) on ice for 30 min. The lysate was then centrifuged at 11,000g for 10 min 4 °C, and the supernatant containing soluble proteins was collected. Protein samples were then separated by SDS–PAGE using gradient gels (4–20%) and transferred onto nitrocellulose membranes for immunoblotting. Membranes were probed with total OXPHOS antibody cocktail (abcam, ab110413), VDAC (abcam, ab14734), POLγA (abcam, ab128899) or tubulin (Sigma, T5168) as figure legend indicated.

### Metabolic flux assay
The experimental method followed the Agilent Seahorse metabolic flux analyser guidelines. In brief, patient fibroblasts depleted of mtDNA and treated with 1 μM PZL-A or vehicle for 7 days were prepared as described previously.

One day prior to the experiment, an XFe96 Pro cartridge was hydrated with 200 μl of calibrant solution per well and left overnight at 37 °C. On the day of the experiment, cells were seeded at $2 × 10^4$ cells per well in DMEM growth medium in a pre-coated Seahorse XFe96 Pro Cell Culture Microplate. The plate was incubated at 37 °C in a $CO_2$ incubator for 2 h to allow cell adhesion.

The growth medium was then replaced with Seahorse Phenol Red-free DMEM supplemented with 5 mM glucose, 2 mM glutamine, and 1 mM sodium pyruvate. Cells were incubated for 1 h at 37 °C to equilibrate in the Seahorse medium. Oxygen consumption rate (OCR) and extracellular acidification rate (ECAR) were measured using the XF Cell Mito stress test kit (Agilent, 103015-100) following the manufacturer's instructions. Specifically, mitochondrial inhibitors (1 μM oligomycin A, 1.5 μM rotenone/antimycin A) or an uncoupler (2 μM FCCP) were diluted in Seahorse medium and loaded into the appropriate ports on the cartridge just before calibration of the Seahorse analyser and subsequent analysis of the cell plate.

Following completion of the Seahorse assay, all medium was removed from the plate and the plate was frozen at −80 °C overnight. For cell quantification, this plate was thawed, and the CyQuant assay (Thermo Fisher, C7026) was performed. Lysis buffer was diluted 1:20 in water and supplemented with CyQuant dye (1:400 dilution). 200 μl of this mixture was added to each well and incubated for 5 min protected from light. Fluorescence was measured using a BMG Labtech Spectrostar plate reader, and cell numbers were quantified using a standard curve.

Data analysis and normalization were conducted using Agilent Seahorse Wave Pro software and Prism 10.

## Statistical analysis

All numerical data are expressed as mean ± s.d. or s.e.m. as stated in the figure legends. Unpaired two-tailed Welch's $t$-test was used to assess statistical significance for comparisons between two groups. For multiple comparisons, Welch and Brown–Forsythe one-way ANOVA was used. Differences were considered statistically significant at $P < 0.05$.

## Ethics statement

The study was conducted according to the Declaration of Helsinki of 1975.

## Reporting summary

Further information on research design is available in the Nature Portfolio Reporting Summary linked to this article.

## Data availability

The atomic models and cryo-EM density maps have been deposited in the Protein Data Bank and the Electron Microscopy Data Bank under the following accession codes: POLγ(G848S)–PZL-A (9GGB, EMD-51326), POLγ(G848S) (9GGC, EMD-51327), POLγ(A467T)–PZL-A (9GGD, EMD-51328), POLγ(A467T) (9GGE, EMD-51329) and wild-type POLγ (9GGF, EMD-51330). PZL-A was assigned a ligand ID (A1IK1) in the Chemical Component Dictionary. Source data are provided with this paper.

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

**Acknowledgements** The work described here was supported by the Swedish Research Council (2021-00932 to M.F., 2022-00976 to C.M.G.), the Swedish Cancer Foundation (to M.F. and C.M.G.), the Knut and Alice Wallenberg Foundation (to M.F. and C.M.G.), and grants from Pretzel Therapeutics. The authors thank the Swedish Biobank, N. Darin and G. Kolberg for help with access to patient fibroblasts; J. P. Uhler for assistance with preparing illustrations; A. Aspegren and E. Korhonen for their help with cell culture work; J. N. Labutti, A. McKenzie and J. J. Novak for valuable discussions during the course of this work; R. Hailes for her valuable comments on the manuscript; U. Alexandersson for purifying some of the proteins used in the study; J. Berndtsson and the Centre for Cellular Imaging at the University of Gothenburg, the National Microscopy Infrastructure (VR-RFI 2019-00217), M. Carroni and K. Walldén at SciLifeLab Stockholm for their assistance with cryo-EM microscopy. The data were collected at the Cryo-EM Swedish National Facility, funded by the Knut and Alice Wallenberg Foundation, the Family Erling Persson Foundation, the Kempe Foundation, SciLifeLab, Stockholm University and Umeå University.

**Author contributions** Conceptualization: C.M.G. and M.F. Defining research goals and experiments: B.M., C.M.G., G.M.B., G.K., J.E.J., M.F., N.-G.L., S.G., T.A.K., X.Z. and X.X. Discovery and development of PZL-A: A.M.G., A.V.G., B.M., B.K.-M., C.G., C.M.G., C.P., C.P.-H., G.K., G.M.B., J.G., J.E.J., J.M.F., L.A., M.F., M.S., P.S.C., S.E., S.G., S.J.K., S.V., T.A.K., V.P., X.X., X.Z. and Y.S. Research supervision: B.M., C.M.G., G.K., J.E.J., J.M.F., M.F., S.G., X.X. and X.Z. Investigation: A.V.G., B.M., C.G., E.H., K.A.S.J., L.J., M.F., M.S., P.S.C., S.G., S.L., S.V., X.Z. and Y.S. Data analysis: A.V.G., B.M., C.G., C.M.G., E.H., K.A.S.J., L.J., M.F., M.S., S.J.K., S.G., S.L., S.V., X.X. and X.Z. Writing, original draft: C.M.G., M.F., S.V. and X.Z. Writing, review and editing: A.M.G., A.V.G., B.M., C.G., C.M.G., C.P., C.P.-H., E.H., G.K., G.M.B., J.G., J.M.F., J.E.J., K.A.S.J., L.A., L.J., M.F., M.S., N.-G.L., P.S.C., S.G., S.J.K., S.L., S.V., T.A.K., V.P., X.S., X.X. and X.Z.

**Funding** Open access funding provided by University of Gothenburg.

**Competing interests** M.F., C.M.G., G.M.B. and N.-G.L. are co-founders of Pretzel Therapeutics. M.F., C.M.G., B.M., X.Z. and N.-G.L. are shareholders of Pretzel Therapeutics and have received consulting fees. S.V. has received consulting fees from Pretzel Therapeutics. M.S., C.G., A.V.G., J.M.F., Y.S., S.J.K., L.A., G.M.B., S.E., C.P.-H., T.A.K., B.K.-M., C.P., J.E.J., X.X. and S.G. are full-time employees of Pretzel Therapeutics and may hold stock or stock options as part of their compensation. V.P. is a former employee of Pretzel Therapeutics. P.S.C., A.M.G., G.K. and J.G. are consultants for Pretzel Therapeutics and hold stock or stock options as part of their compensation. The other authors declare no competing interests.

**Additional information**
**Correspondence and requests for materials** should be addressed to Simon Giroux, Claes M. Gustafsson or Maria Falkenberg.

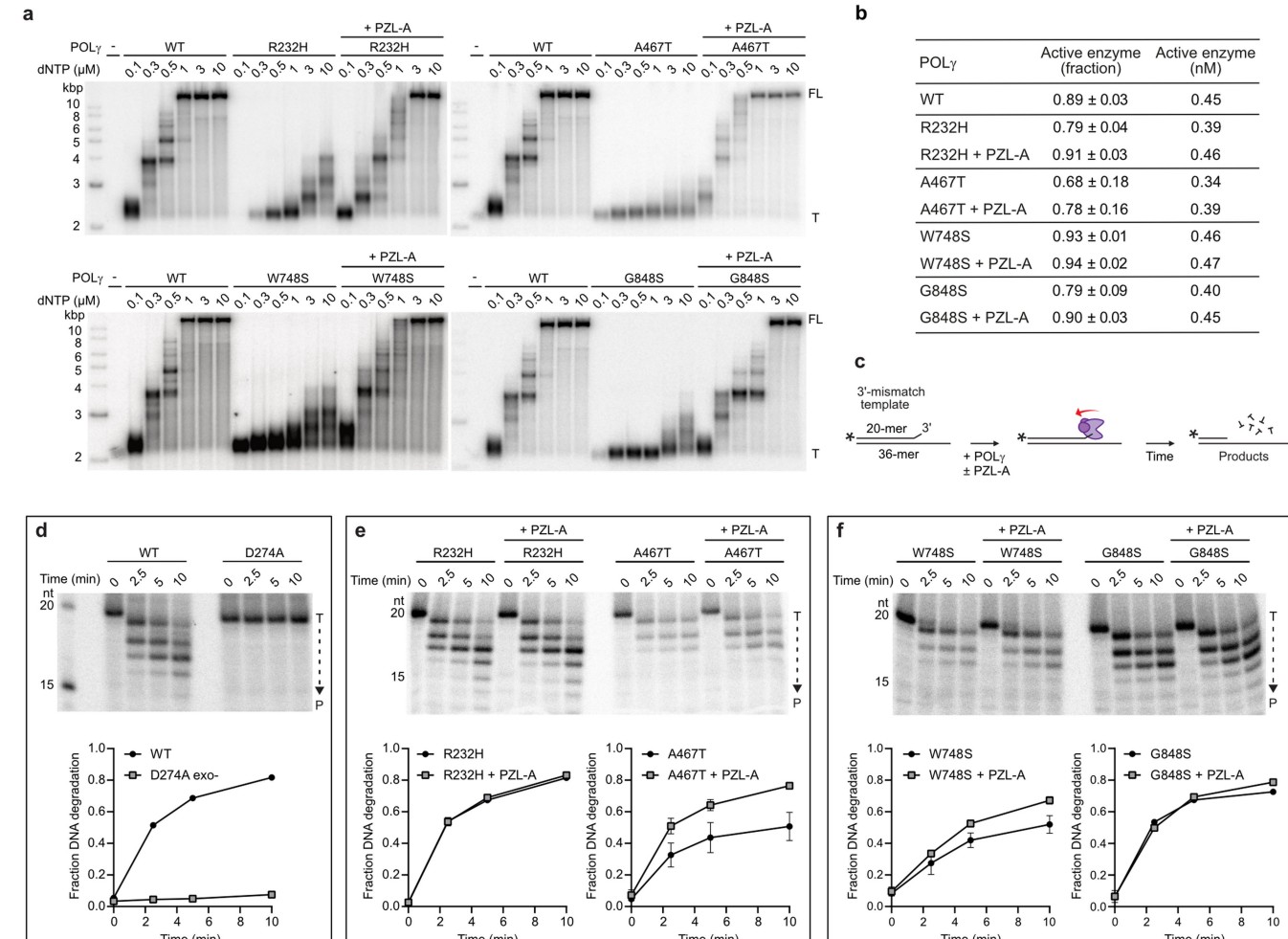

**Extended Data Fig. 1 | Effects of PZL-A on mutant POLγ activities. a**, PZL-A (1 μM) stimulates DNA synthesis in indicated mutant POLγ variants across a broad range of dNTP concentrations. Mutant POLγ variants and mtSSB were incubated with a primed, single-stranded DNA template at 37 °C. Samples were taken at the indicated time points and analyzed on a 0.8% native agarose gel. The positions of the full-length (FL) product and the radiolabelled primer template (T) are indicated. Primers may dissociate from the circular ssDNA during electrophoresis, explaining the lower signals in the template control lanes (-). Representative gels are shown (*n* = 2 independent experiments). **b**, EMSA (electrophoretic mobility shift assay) was used to determine the concentration of active POLγ holoenzyme–DNA complex (active enzyme) under the conditions used in the kinetic experiments (Fig. 1d). The fraction of template-bound POLγ holoenzyme was calculated relative to the unbound template. The concentration was calculated by multiplying the fraction active enzyme by 0.5 nM (maximum theoretical enzyme–DNA complex concentration). Data are presented as the mean ± s.d. (*n* = 3 independent

experiments). **c**, Schematic representation of the exonuclease assay used to investigate effects of PZL-A on the exonuclease activity of POLγ. Drawing by Jennifer Uhler (copyright holder). **d**, Time-course experiments reveal that wild-type POLγ efficiently removes a mismatch at the 3′-end, whereas an exonuclease-deficient, mutant POLγ variant (D274A) is unable to remove the mismatch. POLγ was incubated with the mismatch-template at 37 °C in the absence of dNTPs for times indicated. The products were separated on a 10% urea-PAGE sequencing gel. The positions of products (P) and the radiolabelled mismatch template (T) are indicated. The fraction of short oligonucleotides (shorter than the 19-mer) was quantified and plotted versus time to visualize the DNA degradation. Data are presented as the mean ± s.d. (*n* = 3 independent experiments). **e**–**f**, Time-course experiments demonstrate that PZL-A (1 μM) has no observable negative effect on the exonuclease activity of mutant POLγ variants tested. The experiments were performed as in **d**. Data are presented as the mean ± s.d. (*n* = 3 independent experiments). For gel source data (**a**, **b**, **d**–**f**), see Supplementary Figs. 9–13.

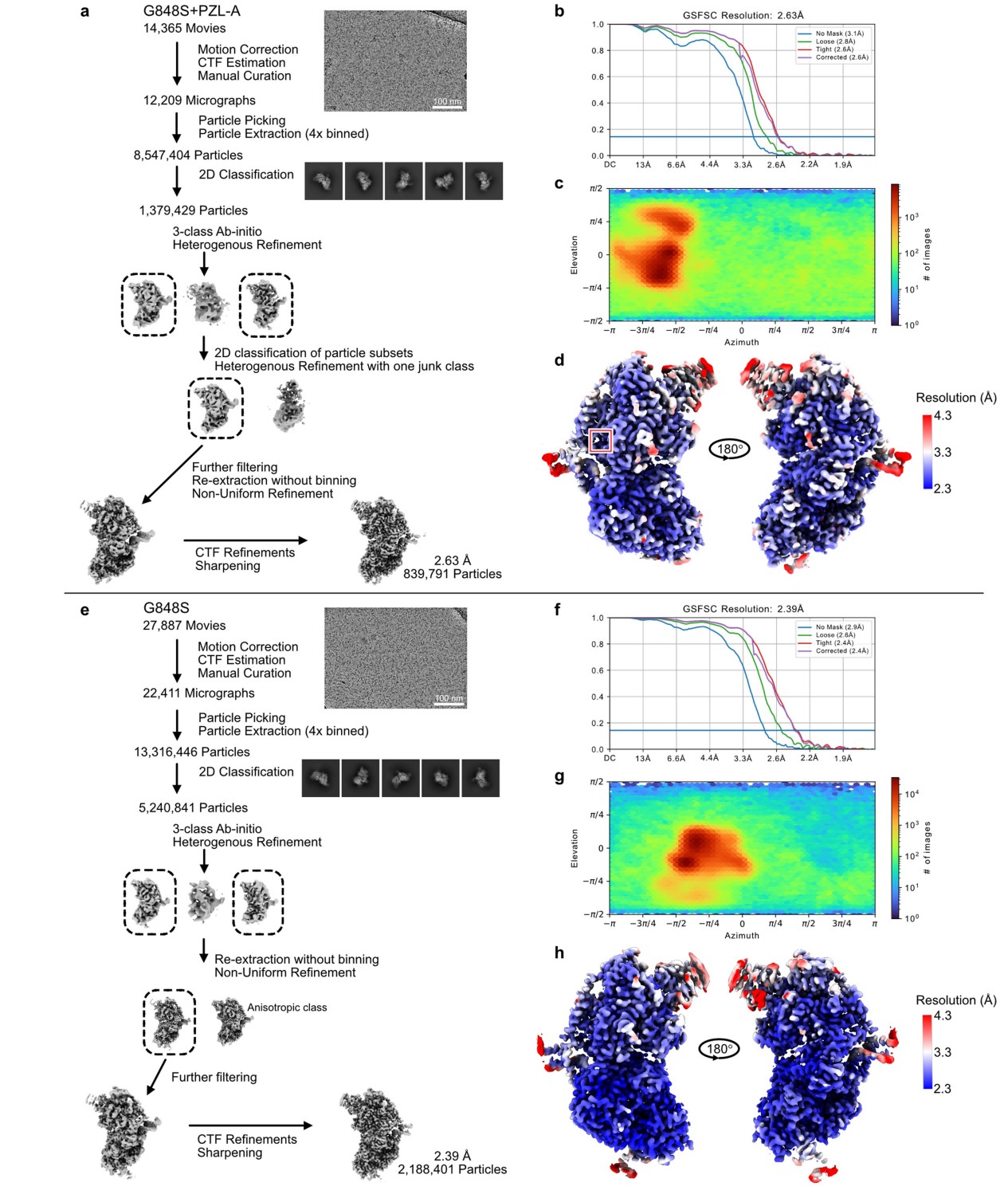

**Extended Data Fig. 2 | Cryo-EM data processing of the G848S-PZL-A and G848S datasets. a**, Cryo-EM processing workflow in cryoSPARC for G848S-PZL-A, with representative images of micrographs and 2D classification averages. **b**, FSC resolution at 0.143 GSFSC threshold as reported by cryoSPARC for the G848S-PZL-A map. **c**, Orientation distribution plot. **d**, Local resolution estimation calculated by cryoSPARC (FSC threshold = 0.5). The position of PZL-A is indicated by the red box. **e**, Cryo-EM processing workflow in cryoSPARC for G848S, with representative images of micrographs and 2D classification averages. **f**, FSC resolution at 0.143 GSFSC threshold as reported by cryoSPARC for the G848S map. **g**, Orientation distribution plot. **h**, Local resolution estimation calculated by cryoSPARC (FSC threshold = 0.5).

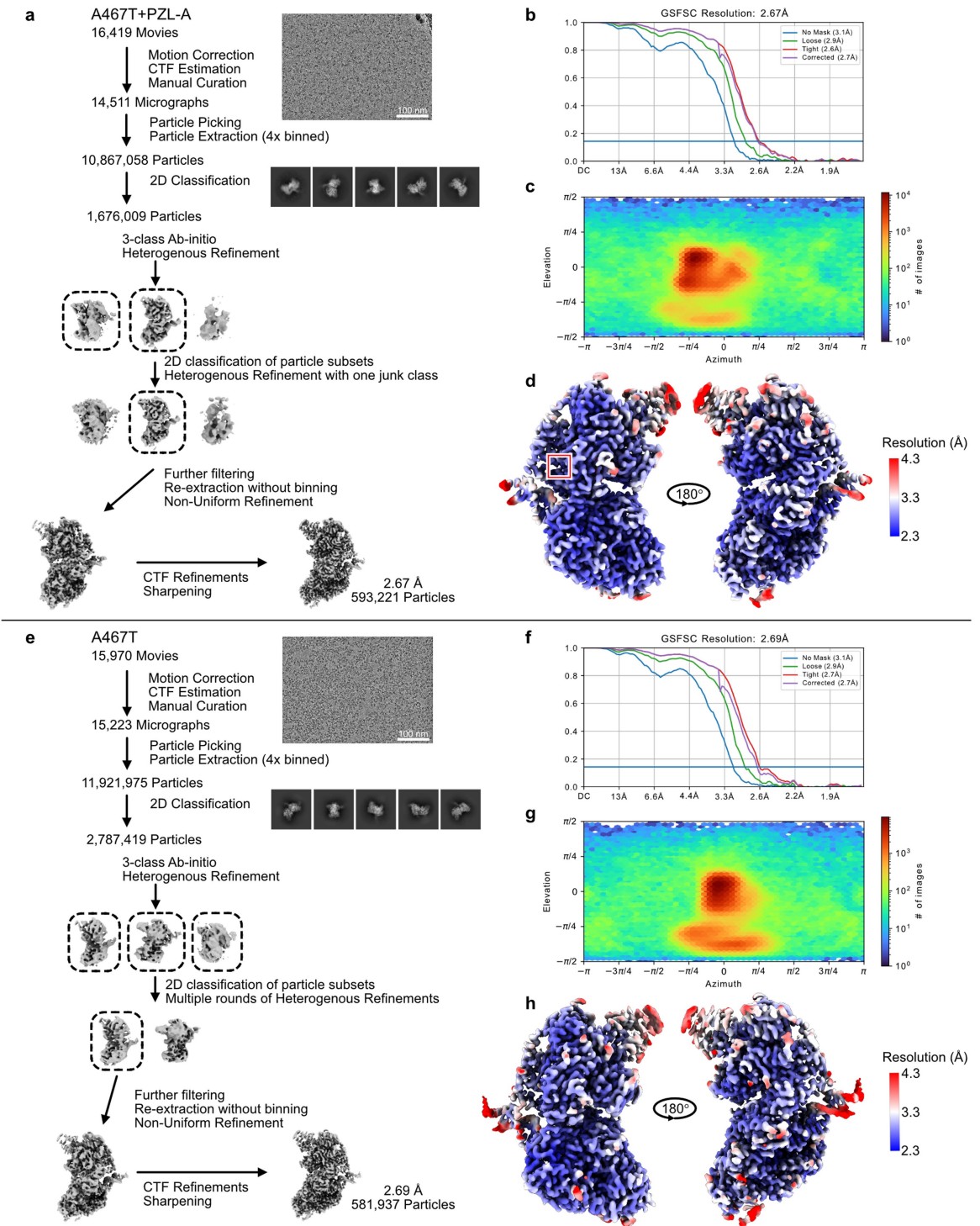

**Extended Data Fig. 3 | Cryo-EM data processing of the A467T-PZL-A and A467T datasets. a**, Cryo-EM processing workflow in cryoSPARC for A467T-PZL-A, with representative images of micrographs and 2D classification averages. **b**, FSC resolution at the 0.143 GSFSC threshold as reported by cryoSPARC for the A467T-PZL-A map. **c**, Orientation distribution plot. **d**, Local resolution estimation calculated by cryoSPARC (FSC threshold = 0.5).

The position of PZL-A is indicated with the red box. **e**, Cryo-EM processing workflow in cryoSPARC for A467T, with representative images of micrographs and 2D classification averages. **f**, FSC resolution at the 0.143 GSFSC threshold as reported by cryoSPARC for the A467T map. **g**, Orientation distribution plot. **h**, Local resolution estimation calculated by cryoSPARC (FSC threshold = 0.5).

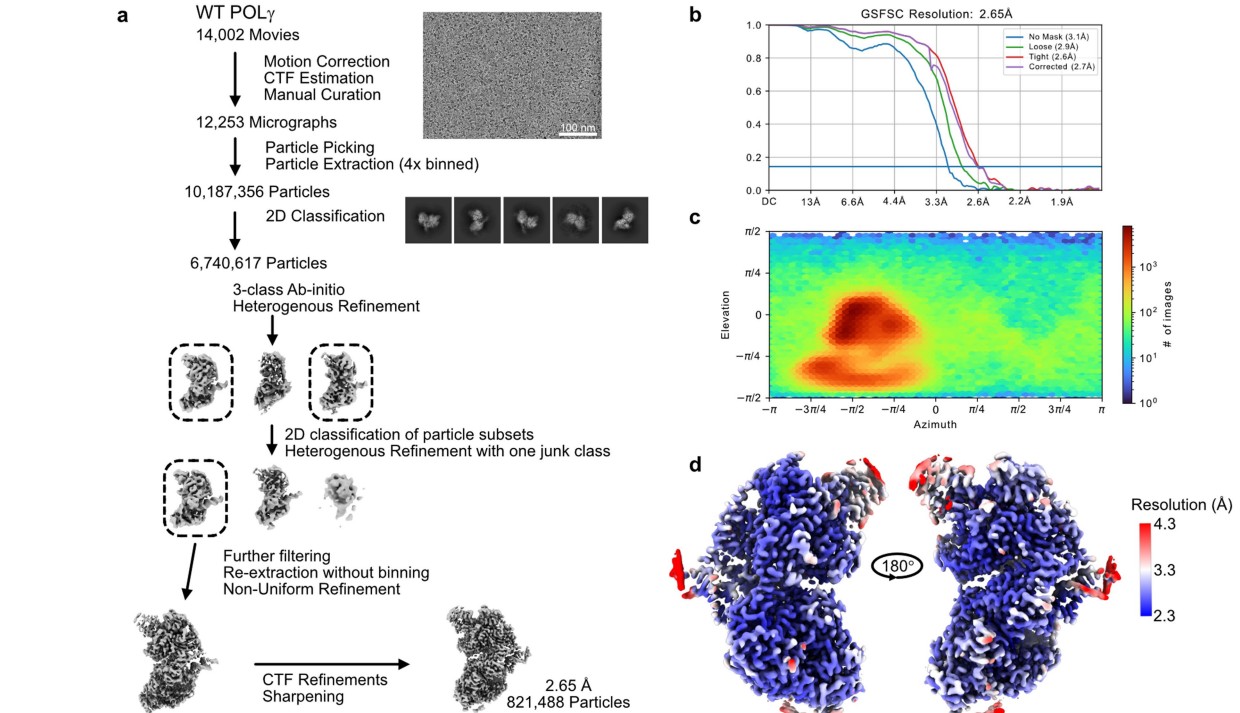

**Extended Data Fig. 4 | Cryo-EM data processing of the wild-type POLγ dataset. a**, Cryo-EM processing workflow in cryoSPARC for wild-type POLγ, with representative images of micrographs and 2D classification averages.

**b**, FSC resolution at the 0.143 GSFSC threshold as reported by cryoSPARC for the wild-type POLγ map. **c**, Orientation distribution plot. **d**, Local resolution estimation calculated by cryoSPARC (FSC threshold = 0.5).

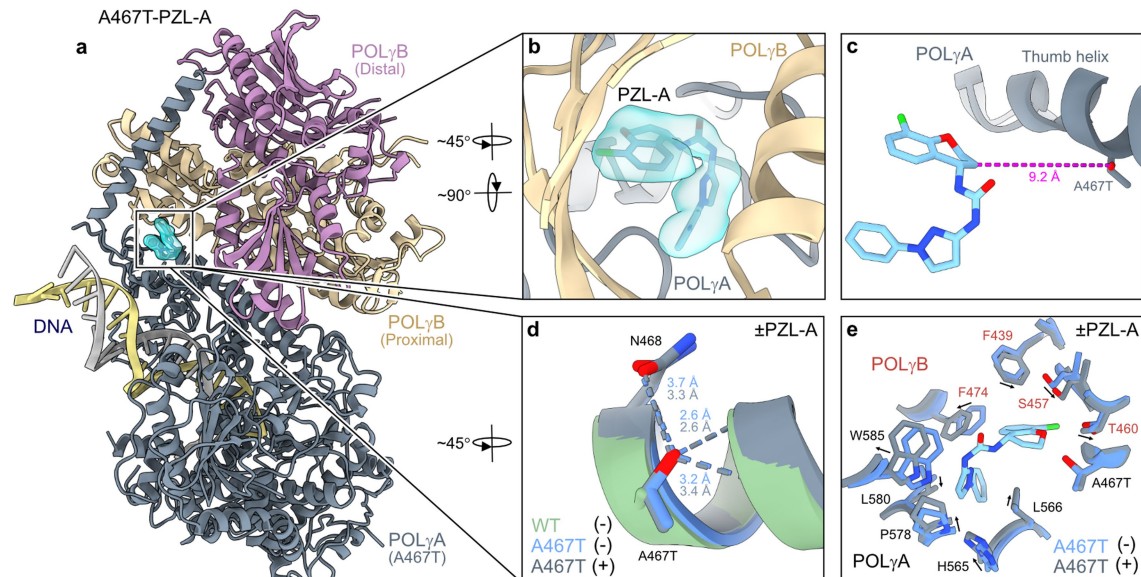

**Extended Data Fig. 5 | Structure of A467T in complex with PZL-A. a**, Cartoon representation of the cryo-EM structure of A467T bound to PZL-A. The binding pocket is highlighted with the cryo-EM density for PZL-A (cyan). PZL-A binds POLγ in a pocket at the interface between POLγA and the proximal POLγB subunit. **b**, Close-up of PZL-A and its cryo-EM density in the binding pocket. **c**, PZL-A binds approximately 9.2 Å from the A467T residue, which resides in one of the thumb helices. **d**, A467T (blue) and A467T-PZL-A (grey) structures overlaid on the wild-type POLγ (green) structure at the mutation location. Potential hydrogen bonds are shown as dashed lines. **e**, Superimposition of the A467T (blue) and A467T-PZL-A (grey) structures at the binding site of PZL-A. Minor positional changes of nearby residues can be observed upon binding of PZL-A, and the directions of these adjustments are shown with arrows. A467T is not affected by PZL-A binding.

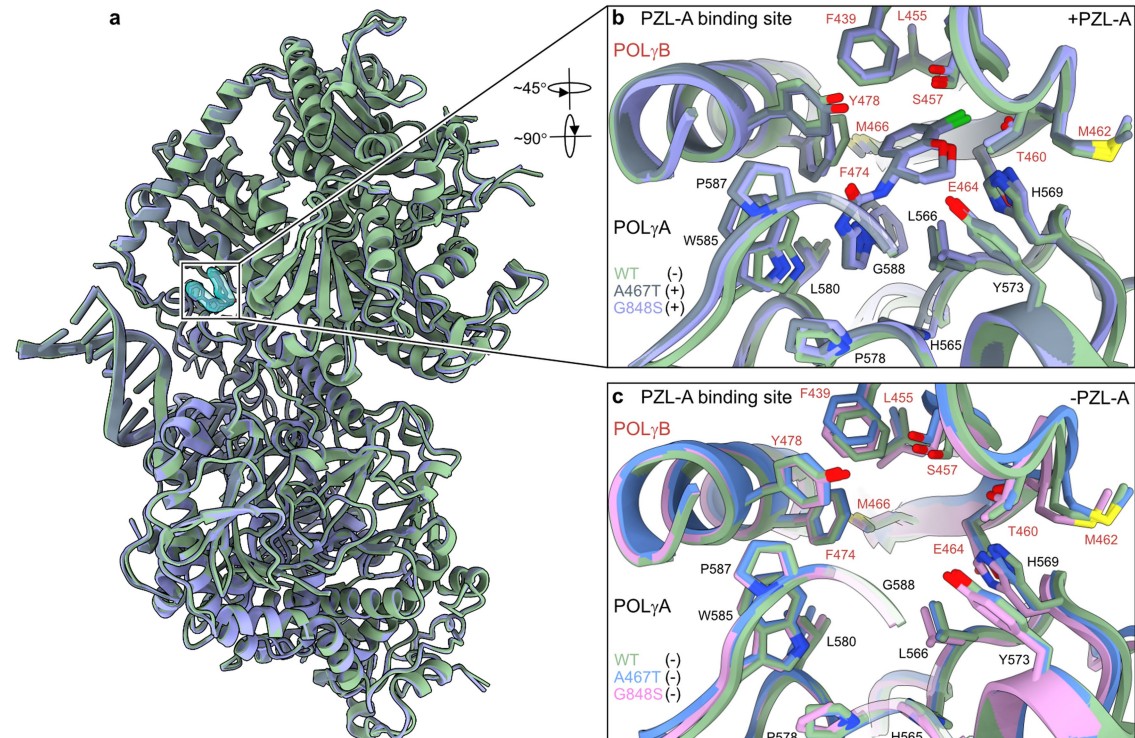

**Extended Data Fig. 6 | Structural comparison of the PZL-A binding pocket in wild-type and mutant POLγ. a**, Overview of overlaid A467T-PZL-A (grey), G848S-PZL-A (purple), and wild-type POLγ (green) structures. **b**, Close-up of the PZL-A binding pocket in the overlaid structures. Sidechains are shown as sticks and labelled in black (POLγA) or brown (POLγB). **c**, Close-up of the empty PZL-A binding pocket in the A467T (blue), G848S (pink), and wild-type POLγ (green) apo-structures. Sidechains are shown as sticks and labelled in black (POLγA) or brown (POLγB).

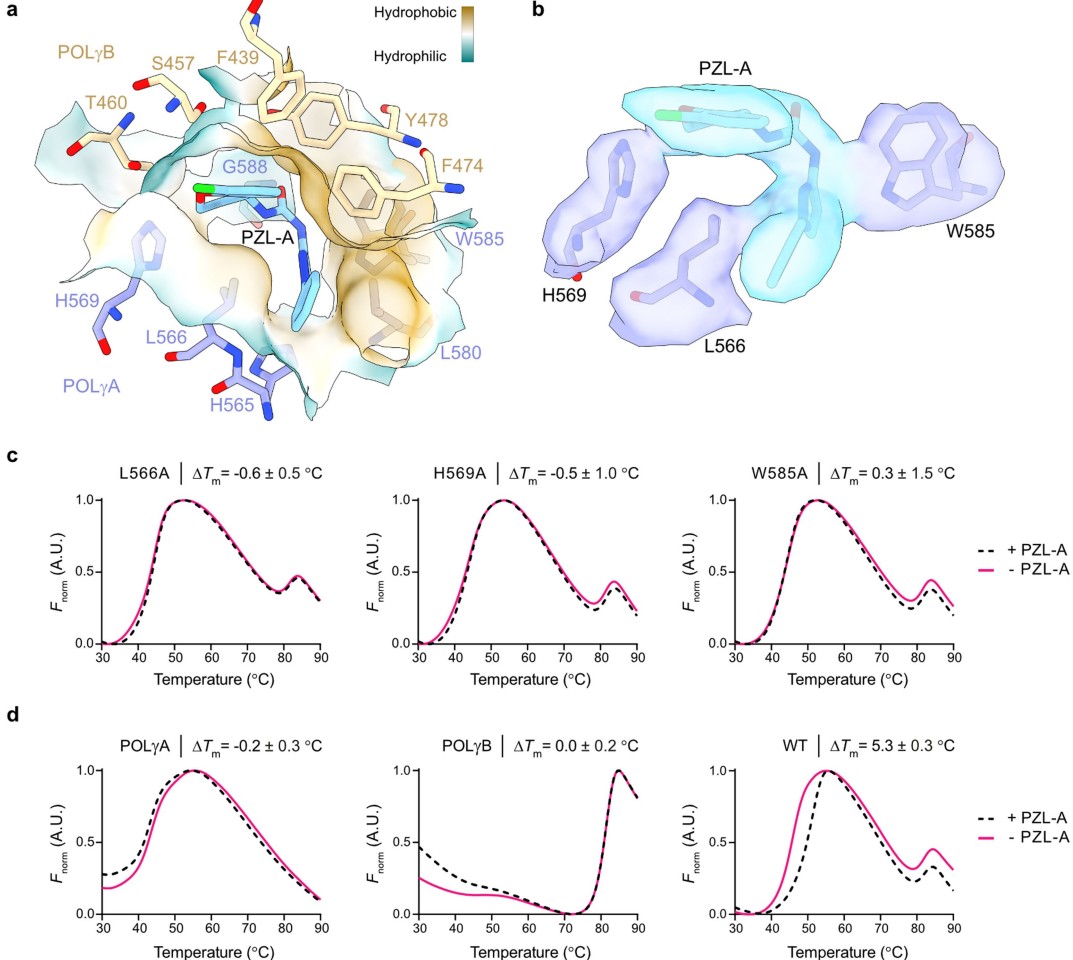

**Extended Data Fig. 7 | Mutagenesis of the PZL-A binding pocket. a**, PZL-A in the binding site in the G848S-PZL-A structure, with sidechains from nearby residues displayed as sticks. The protein surface surrounding the compound is coloured by hydrophobicity potential (calculated in UCSF ChimeraX). The binding site is mainly a hydrophobic pocket, with PZL-A wedged between hydrophobic residues from both POLγA (light purple) and POLγB (beige). However, a small hydrophilic patch is located near the polar urea carbonyl group and the chromane oxygen in PZL-A. **b**, Three residues in POLγA (L566, H569 and W585) are in close proximity to PZL-A and interacts with different parts of the compound. Cryo-EM densities for PZL-A (blue) and the residues (purple) are shown. **c**, Representative plots of differential scanning fluorimetry performed with POLγA variants in complex with POLγB, in the absence or presence of 10 μM PZL-A. Residues in close proximity to PZL-A (see **b**) were replaced with alanine (L566A, H569A and W585A). Values are presented as normalized fluorescence ($F_{norm}$) in arbitrary units (A.U.). The difference in melting temperature, $\Delta T_m$ (mean ± s.d., $n$ = 3 independent experiments) is given. **d**, Representative plots of differential scanning fluorimetry performed with wild-type POLγA, wild-type POLγB, and the wild-type POLγ complex, in the absence or presence of 10 μM PZL-A. Values are presented as normalized fluorescence ($F_{norm}$) in arbitrary units (A.U.). The difference in melting temperature, $\Delta T_m$ (mean ± s.d., $n$ = 3 independent experiments) is given. For the wild-type POLγ complex, $\Delta T_m$ is given for the POLγA subunit.

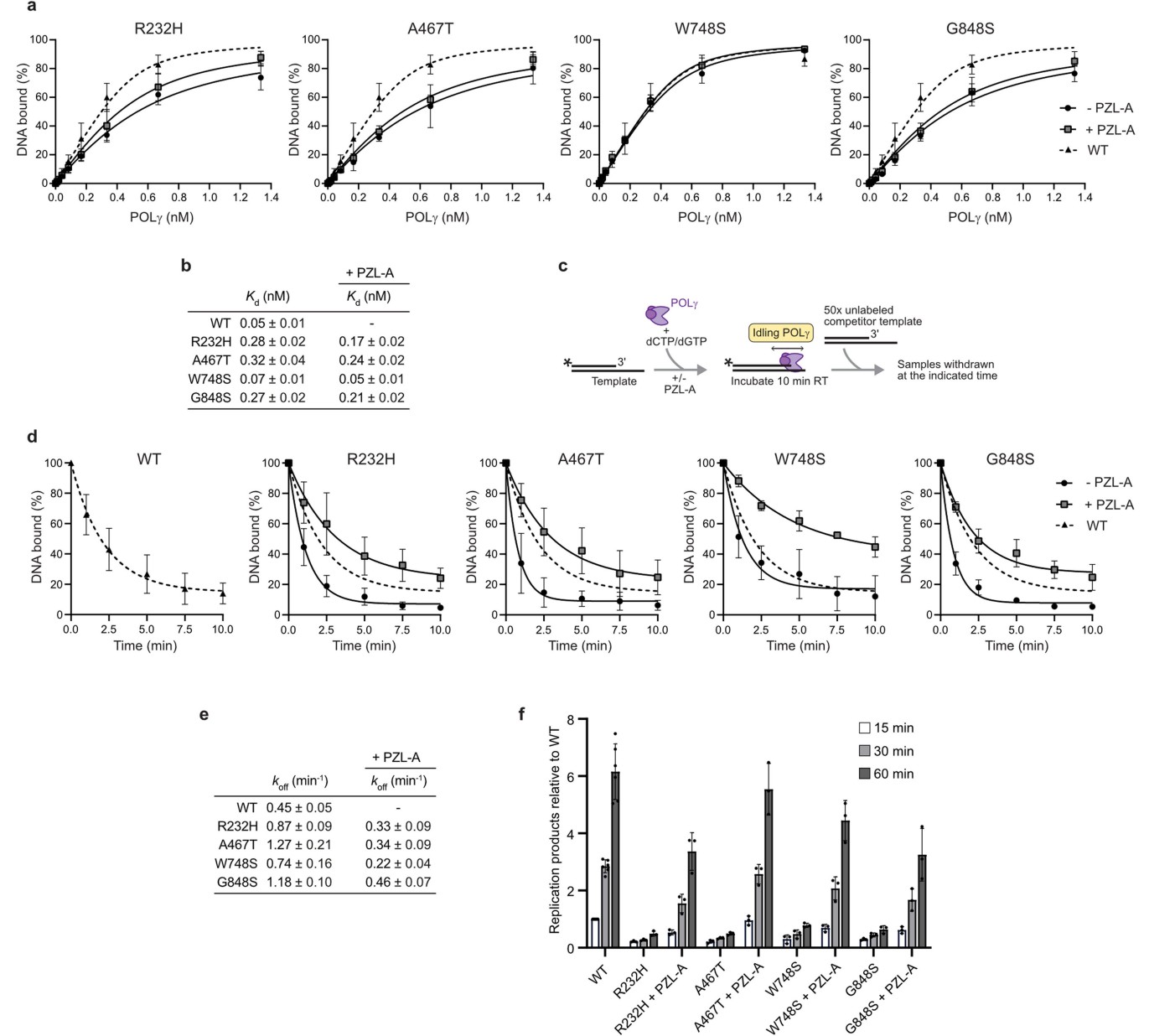

**Extended Data Fig. 8 | PZL-A stabilizes mutant POLγ interactions with the template during active DNA synthesis. a**, Formation of stalled elongation complexes was analysed by EMSA in the absence or presence of PZL-A. Complex formation was plotted against POLγ concentrations, and the binding curves were fitted to a quadratic equation for tight-binding to determine the dissociation constants ($K_d$). Presented data are the mean ± s.d. ($n = 3$ independent experiments). **b**, Summary of the determined dissociation constants, $K_d$ (nM) from experiments in **a**. Presented data are the mean ± s.e.m. ($n = 3$ independent experiments). **c**, Schematic representation of the competition experiment used to evaluate DNA binding of idling POLγ. The indicated POLγ variants were incubated with a radiolabelled template at room temperature in the presence of dCTP and dGTP. A 50-fold excess of unlabelled competitor template was added, and the reactions were incubated at 37 °C for the indicated times. Samples were separated using 4% native-PAGE, and the fraction of the formed complex was quantified and plotted against time to determine the dissociation rate constant. Drawing by Jennifer Uhler (copyright holder). **d**, The percentage of labeled DNA bound to POLγ during idling was plotted versus time (minutes after addition of unlabeled DNA) and the data were fit to an exponential dissociation model to determine the dissociation rate constant, $k_{off}$. The fit to the wild-type data (dashed line) was added to the mutant plots for comparison. Data points are mean ± s.d. ($n = 3$ independent experiments). **e**, Summary of the determined dissociation rate constants, $k_{off}$ ($min^{-1}$) from experiments described in **c**,**d**. Presented data are the mean ± s.e.m. ($n = 3$ independent experiments). **f**, Quantification of the rolling circle assay in Fig. 3e. Data are presented as replication products relative to WT (15 min time point). The values from three timepoints (15, 30 and 60 min) were plotted for each mutant ± PZL-A as well as wild-type. Data presented are mean ± s.d., $n = 3$ (mutants) or $n = 6$ (wild-type) independent experiments. For gel source data (**a**, **b**, **d**, **e**), see Supplementary Figs. 14–16.

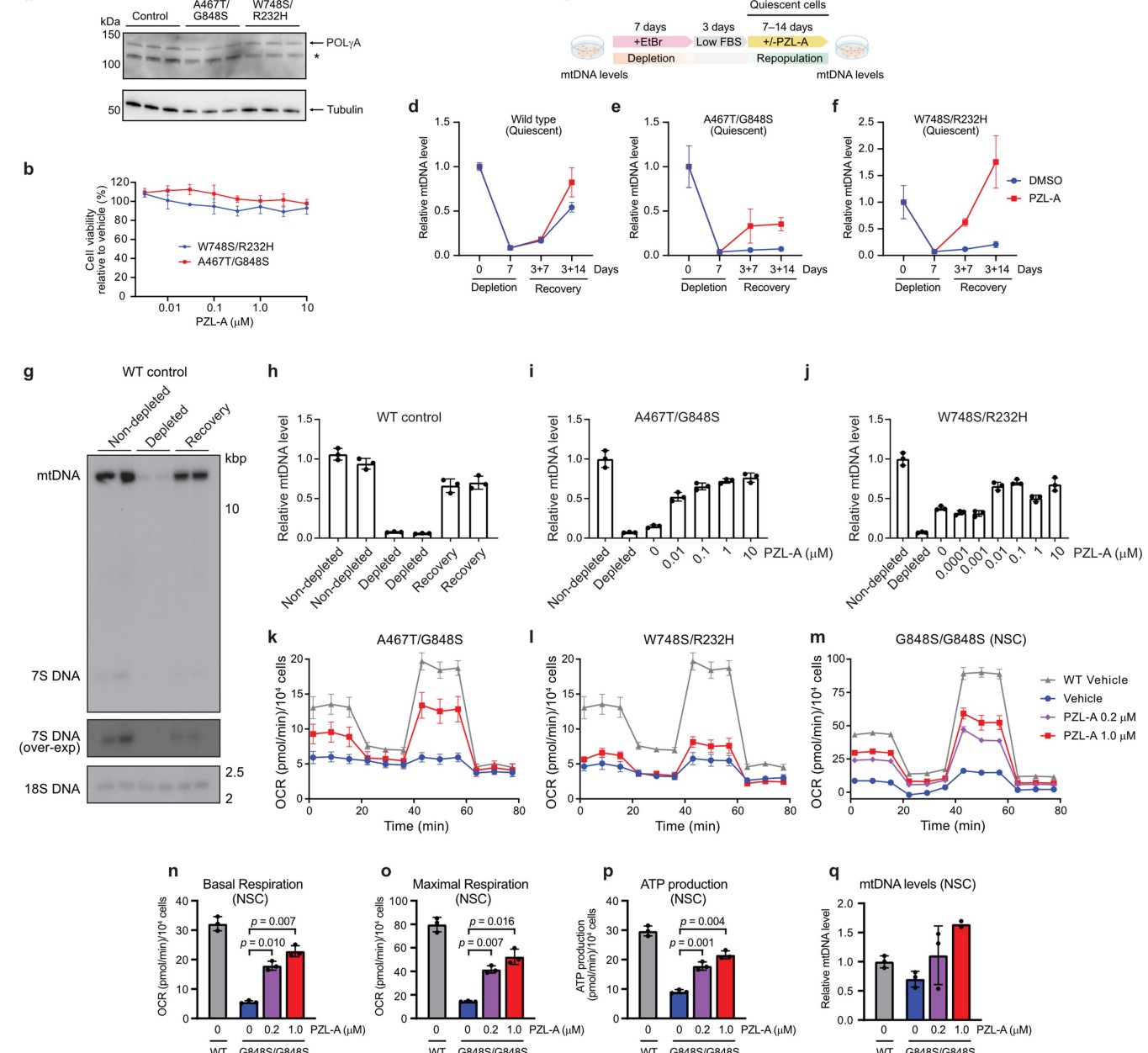

**Extended Data Fig. 9 | PZL-A stimulates mtDNA synthesis and ATP production. a**, Immunoblot analysis of POLγA in control and mutant fibroblasts. Tubulin was used as a loading control. *indicates a non-specific band. Representative blot is shown (*n* = 2 independent experiments). **b**, Cell viability of A467T/G848S, W748S/R232H fibroblasts after PZL-A treatment for 5 days at indicated concentrations relative to vehicle treatment (mean ± s.d., *n* = 3 biological replicates). **c**, Schematic illustration of the mtDNA depletion and recovery experiment in quiescent cells. Drawing by Jennifer Uhler (copyright holder). **d–f**, Depletion of mtDNA followed by repopulation in quiescent fibroblasts (wild-type or indicated mutant cells) after 7 days of treatment with 50 ng/ml EtBr. After 7 days of EtBr treatment, the cells reached confluence and FBS was reduced to 0.1% to induce quiescence. PZL-A and vehicle treatment were started after removal of EtBr and DNA samples were analyzed at the indicated time points (mean ± s.d., *n* = 3 biological replicates). **g**, The mtDNA levels in wild-type fibroblasts after EtBr depletion (50 ng/ml) for 7 days, followed by recovery for 10 days. Southern blot analysis after BamHI digestion was used to detect mtDNA and 7S DNA. Nuclear 18S rDNA is used as a loading control. **h–j**, Relative mtDNA levels were assessed with real-time quantitative PCR (qPCR). Samples analyzed correspond to those presented in

Extended Data Fig. 9g and Fig. 4e,f respectively. (mean ± s.d., *n* = 3 technical replicates). **k–l**, Bioenergetic profiles of 1 μM PZL-A or vehicle treated mutant and wild-type fibroblasts were measured by Seahorse XFe96 pro extracellular flux analyzer and used to calculate parameters reported in Fig. 4i–k (mean ± s. e.m. *n* = 3 biological replicates). **m**, Bioenergetic profiles of 0.2 and 1 μM PZL-A or vehicle treated G848S/G848S mutant and PGP-1 wild-type NSCs were measured by Seahorse XFe96 pro extracellular flux analyzer (mean ± s.e.m. *n* = 3 biological replicates). **n–p**, Mitochondrial respiration changes in G848S/G848S mutant and wild-type NSCs treated with PZL-A or vehicle. A Seahorse XFe96 pro extracellular flux analyzer was used to analyze basal respiration, maximal respiration, and ATP production. The cells were treated for 10 days before transferring to Seahorse XFe96 Pro plates. Data presented are the mean ± s.d. (*n* = 3 biological replicates). Welch and Brown-Forsythe one-way ANOVA was used determine all p-values. **q**, Relative mtDNA levels were assessed in G848S/G848S mutant and wild-type NSCs treated with PZL-A or vehicle (mean ± s.d., *n* = 3 biological replicates except *n* = 2 in 1 μM PZL-A treatments). Samples analyzed correspond to those presented in **m–p**. For gel source data (**a**, **g**), see Supplementary Fig. 17.

**Extended Data Table 1 | Cryo-EM data collection, refinement and validation statistics**

| | G848S-PZL-A (EMDB-51326) (PDB 9GGB) | G848S (EMDB-51327) (PDB 9GGC) | A467T-PZL-A (EMDB-51328) (PDB 9GGD) | A467T (EMDB-51329) (PDB 9GGE) | WT (EMDB-51330) (PDB 9GGF) |
|---|---|---|---|---|---|
| **Data collection and processing** | | | | | |
| Magnification | 105,000 | 105,000 | 105,000 | 105,000 | 105,000 |
| Voltage (kV) | 300 | 300 | 300 | 300 | 300 |
| Electron exposure (e⁻/Å²) | 40 | 40 | 40 | 40 | 40 |
| Defocus range ($\mu$m) | 0.8-2.2 | 0.8-2.2 | 0.8-2.2 | 0.8-2.2 | 0.8-2.2 |
| Pixel size (Å) | 0.825 | 0.828 | 0.825 | 0.828 | 0.825 |
| Symmetry imposed | C1 | C1 | C1 | C1 | C1 |
| Initial particle images (no.) | 8,547,404 | 13,316,446 | 10,867,058 | 11,921,975 | 10,187,356 |
| Final particle images (no.) | 893,791 | 2,188,401 | 593,221 | 581,937 | 821,488 |
| Map resolution (Å) | 2.63 | 2.39 | 2.67 | 2.69 | 2.65 |
| FSC threshold | 0.143 | 0.143 | 0.143 | 0.143 | 0.143 |
| Map resolution range (Å) | 2.3-42.1 | 2.1-23.0 | 2.3-25.5 | 2.3-34.7 | 2.3-34.2 |
| | | | | | |
| **Refinement** | | | | | |
| Initial model used (PDB code) | 4ZTZ | 4ZTZ | 4ZTZ | 4ZTZ | 4ZTZ |
| Model resolution (Å) | 3.1 | 2.8 | 3.0 | 3.1 | 3.0 |
| FSC threshold | 0.5 | 0.5 | 0.5 | 0.5 | 0.5 |
| Model composition | | | | | |
| Non-hydrogen atoms | 13,935 | 13,930 | 14,056 | 13,855 | 13,925 |
| Protein/DNA residues | 1,635/38 | 1,638/38 | 1,650/38 | 1,628/38 | 1,637/38 |
| Ligands | 3 | 2 | 3 | 2 | 2 |
| *B* factors (Å²) | | | | | |
| Protein | 52.56 | 44.53 | 46.16 | 48.30 | 44.39 |
| DNA | 68.43 | 59.94 | 58.64 | 62.35 | 59.45 |
| Ligands | 47.37 | 33.81 | 41.29 | 42.50 | 39.97 |
| R.m.s. deviations | | | | | |
| Bond lengths (Å) | 0.004 | 0.002 | 0.004 | 0.002 | 0.002 |
| Bond angles (°) | 0.524 | 0.429 | 0.522 | 0.433 | 0.444 |
| Validation | | | | | |
| MolProbity score | 1.51 | 1.35 | 1.41 | 1.36 | 1.30 |
| Clashscore | 7.48 | 6.41 | 7.48 | 6.44 | 5.50 |
| Poor rotamers (%) | 0.71 | 0.28 | 0.56 | 0.57 | 0.89 |
| Ramachandran plot | | | | | |
| Favored (%) | 97.51 | 98.39 | 98.15 | 98.31 | 98.51 |
| Allowed (%) | 2.49 | 1.61 | 1.85 | 1.69 | 1.49 |
| Disallowed (%) | 0.00 | 0.00 | 0.00 | 0.00 | 0.00 |

Claes Gustafsson
Maria Falkenberg Gustafsson

# Reporting Summary

## Statistics

For all statistical analyses, confirm that the following items are present in the figure legend, table legend, main text, or Methods section.

| n/a | Confirmed | |
|---|---|---|
| ☐ | ☒ | The exact sample size (*n*) for each experimental group/condition, given as a discrete number and unit of measurement |
| ☐ | ☒ | A statement on whether measurements were taken from distinct samples or whether the same sample was measured repeatedly |
| ☐ | ☒ | The statistical test(s) used AND whether they are one- or two-sided *Only common tests should be described solely by name; describe more complex techniques in the Methods section.* |
| ☒ | ☐ | A description of all covariates tested |
| ☐ | ☒ | A description of any assumptions or corrections, such as tests of normality and adjustment for multiple comparisons |
| ☐ | ☒ | A full description of the statistical parameters including central tendency (e.g. means) or other basic estimates (e.g. regression coefficient) AND variation (e.g. standard deviation) or associated estimates of uncertainty (e.g. confidence intervals) |
| ☐ | ☒ | For null hypothesis testing, the test statistic (e.g. *F*, *t*, *r*) with confidence intervals, effect sizes, degrees of freedom and *P* value noted *Give P values as exact values whenever suitable.* |
| ☒ | ☐ | For Bayesian analysis, information on the choice of priors and Markov chain Monte Carlo settings |
| ☒ | ☐ | For hierarchical and complex designs, identification of the appropriate level for tests and full reporting of outcomes |
| ☒ | ☐ | Estimates of effect sizes (e.g. Cohen's *d*, Pearson's *r*), indicating how they were calculated |

*Our web collection on statistics for biologists contains articles on many of the points above.*

## Software and code

Policy information about availability of computer code

| Data collection | BMG PHERAstar microtiter plate reader  control software(v5.70 R6); BioRad CFX Maestro real time software(v2.3); Agilent Seahorse Wave Pro Software(v2.6) |
|---|---|
| Data analysis | In addition to above mentioned software,the following  softwares were used in this study: BMG MARS data analysis software(V4.01 R2); Fujifilm Multi Gauge software(V3.1);  BioRad Image lab software(v6.1);cryoSPARC (v4.3.1);DeepEMhancer(v0.14); Coot (v.0.9.8.1) and ISOLDE (v.1.4);PHENIX (v.1.20.);UCSF ChimeraX(v1.4); Graphpad Prism Software (v8.0 and 10.0);MolProbity(v4.02b-467) |

For manuscripts utilizing custom algorithms or software that are central to the research but not yet described in published literature, software must be made available to editors and reviewers. We strongly encourage code deposition in a community repository (e.g. GitHub). See the Nature Portfolio guidelines for submitting code & software for further information.

## Data

Policy information about availability of data

All manuscripts must include a data availability statement. This statement should provide the following information, where applicable:
- Accession codes, unique identifiers, or web links for publicly available datasets
- A description of any restrictions on data availability
- For clinical datasets or third party data, please ensure that the statement adheres to our policy

All relevant data generated and analyzed in this study are available online. Uncropped gels and raw data are provided in the supplementary information and source

data. For POLG mutations the human DNA Polymerase Gamma Mutation Database was used (https://tools.niehs.nih.gov/polg/). The atomic models and cryo-EM density maps have been deposited in the Protein Data Bank and the Electron Microscopy Data Bank under the following accession codes: G848S-PZL-A (9GGB, EMD-51326), G848S (9GGC, EMD-51327), A467T-PZL-A (9GGD, EMD-51328), A467T (9GGE, EMD-51329) and wild-type POLg (9GGF, EMD-51330). PZL-A was assigned a ligand ID (A1IK1) in the Chemical Component Dictionary.

# Research involving human participants, their data, or biological material

Policy information about studies with human participants or human data. See also policy information about sex, gender (identity/presentation), and sexual orientation and race, ethnicity and racism.

| | |
|---|---|
| Reporting on sex and gender | No sex and gender were applied in this study. |
| Reporting on race, ethnicity, or other socially relevant groupings | No race, ethnicity and socially relevant groupings were applied in this study. |
| Population characteristics | N/A |
| Recruitment | N/A |
| Ethics oversight | N/A |

Note that full information on the approval of the study protocol must also be provided in the manuscript.

# Field-specific reporting

Please select the one below that is the best fit for your research. If you are not sure, read the appropriate sections before making your selection.

☒ Life sciences        ☐ Behavioural & social sciences        ☐ Ecological, evolutionary & environmental sciences

For a reference copy of the document with all sections, see [nature.com/documents/nr-reporting-summary-flat.pdf](nature.com/documents/nr-reporting-summary-flat.pdf)

# Life sciences study design

All studies must disclose on these points even when the disclosure is negative.

| | |
|---|---|
| Sample size | We use a sample size of n=3 in our cellular and biochemical experiments as it aligns with standard practices in the field, providing sufficient replicates to assess experimental reproducibility while balancing feasibility, resource constraints, and biological variability. |
| Data exclusions | No samples were excluded |
| Replication | All experimental data was reliably reproduced as indicated in the method section and figure legends. |
| Randomization | Randomization was not applied in this study because our experiments were based on biochemical and cellular assays rather than subject allocation or treatment assignment. All experimental conditions were tightly controlled by using standardized protocols, including identical cell culture conditions, reagent batches, and assay parameters, thereby minimizing potential sources of variability. |
| Blinding | Investigators were not blinded to experimental design and outcome. Investigators were also not blinded during data analysis. Blinding was not applied in this study because the experiments were based on biochemical and cellular assays, where measurements were obtained using objective, quantitative methods such as fluorescence-based enzymatic assays, qPCR, and cryo-EM structural analysis. These techniques provide direct and unbiased readouts, minimizing the risk of subjective bias in data interpretation. |

# Reporting for specific materials, systems and methods

We require information from authors about some types of materials, experimental systems and methods used in many studies. Here, indicate whether each material, system or method listed is relevant to your study. If you are not sure if a list item applies to your research, read the appropriate section before selecting a response.

## Materials & experimental systems

| n/a | Involved in the study |
|-----|----------------------|
| ☐ | ☒ Antibodies |
| ☐ | ☒ Eukaryotic cell lines |
| ☒ | ☐ Palaeontology and archaeology |
| ☒ | ☐ Animals and other organisms |
| ☒ | ☐ Clinical data |
| ☒ | ☐ Dual use research of concern |
| ☒ | ☐ Plants |

## Methods

| n/a | Involved in the study |
|-----|----------------------|
| ☒ | ☐ ChIP-seq |
| ☒ | ☐ Flow cytometry |
| ☒ | ☐ MRI-based neuroimaging |

# Antibodies

| | |
|---|---|
| Antibodies used | Total OXPHOS Rodent WB Antibody Cocktail (ab110413, Abcam); anti-tubulin (T5168,Sigma); anti-VDAC(ab14734,abcam) Anti-POLG antibody(ab128899,abcam) |
| Validation | - Total OXPHOS Rodent WB Antibody Cocktail (ab110413, Abcam): The Abpromise covers the use of the antibody for WB application. The antibody has been referenced in 1205 publications. https://www.abcam.com/total-oxphos-rodent-wb-antibody- cocktail-ab110413.html<br>- anti-tubulin (T5168, Sigma): The antibody was subjected to enhanced antibody validation and referenced in 4999 publications base on citeab.com. https://www.sigmaaldrich.com/SE/en/product/sigma/t5168?srsltid=AfmBOooIvGCftVFBNQWHaoTNKBwwUzXEvbJvMulIqanrZHxRaap_z90i https://www.citeab.com/antibodies/2304940-t5168-monoclonal-anti-tubulin-antibody-produced-in<br>-anti-VDAC (ab14734,abcam): The Abpromise covers the use of the antibody for WB application. The antibody has been referenced in 555 publications. https://www.abcam.com/en-us/products/primary-antibodies/vdac1-porin-vdac3-antibody-20b12af2-ab14734<br>Anti-POLG antibody(ab128899,abcam) Rabbit Monoclonal POLG antibody. Suitable for WB and reacts with Mouse, Rat, Human samples. Cited in 28 publications. https://www.citeab.com/antibodies/761776-ab128899-recombinant-anti-polg-antibody-epr7296?des=c7bdf0f22e10ee55 |

# Eukaryotic cell lines

Policy information about cell lines and Sex and Gender in Research

| | |
|---|---|
| Cell line source(s) | Human skin biopsy derived primary fibroblasts were obtained from the Swedish Biobank. |
| Authentication | All cell lines used in this study have been verified by Eurofins Genomics STR analysis of human cell lines. |
| Mycoplasma contamination | Cell lines were routinely examined for mycoplasma contamination (negative). |
| Commonly misidentified lines (See ICLAC register) | None |

# Plants

| | |
|---|---|
| Seed stocks | N/A |
| Novel plant genotypes | N/A |
| Authentication | N/A |

