## [Peer Review File · Nature]

SMALL MOLECULES RESTORE MUTANT MITOCHONDRIAL DNA POLYMERASE ACTIVITY

Corresponding Author: Professor Maria Falkenberg

This file contains all reviewer reports in order by version, followed by all author rebuttals in order by version. Parts of this Peer Review File have been redacted as indicated to maintain the confidentiality of unpublished data.

Version 0:

Reviewer comments:

Referee #1

(Remarks to the Author)

The paper by Valenzuela et al. is a breakthrough piece of work towards the identification of rational, experimentally validated therapeutic approaches in POLG mutations, one of the major causes of mitochondrial disorders in humans. POLG displays a marked heterogeneity in its pathogenic mutations, associated with comparable clinical and pathological diversity. The aim of the work was to find a general tool able to stabilize and increase the proficiency of the enzyme in various POLG mutations, which could thus be used in a large set of pathogenic variants. The work was based first on the evaluation of 270000 compounds to increase the efficiency of an high-throughput in vitro DNA synthesis assay operated by wt POLG and POLG carrying two relatively frequent human polgamma-A mutations, the A467T and the G848S amino acid changes. One single compound (compound 1) was identified, which was further modified to increase its stimulatory activity, leading to a more potent molecule called PZL-A. PZL-A displayed an effect in the nanomolar range (AC50 20-200 nM), which determined a recovery of DNA synthesis in the mutant POLG variants nearly identical to the wild type. This very encouraging result was associated with increase of V_{max} and k_{cat} , with a more modest effect on K_m . This is an extremely relevant result which for the first time demonstrates that the low nucleotide incorporation activity of hypomorphic variants of POLG can be effectively corrected by the action of a small molecule. The lack of effect on K_m can perhaps be attributed to the fact that the mutations used in the study do not affect directly the catalytic site, but it would be interesting to know the authors' opinion. Likewise, no effect of the PZL-A was recorded for the 3'-5' exonuclease activity of the POLG protein that serves as a proofreading tool during mtDNA replication. Next, using cryo-EM technology, PZL-A was localized in a hydrophobic pouch of the polgamma-A protein in close proximity with the polgamma-B protein that participates in the formation of the mature heterotrimeric enzyme. The authors showed that the addition of PZL-A increases the thermal stability in the mutant POLG proteins together with a clear increase of the polymerase activity compared to the wt. The results are really impressive, showing a remarkable increase of dNTP incorporation reaching 60-80% of the wt values. The cryo-EM studies do prove that the PZL-A compound, which forms a C-like structure, is indeed specifically associated with the mentioned hydrophobic region and can establish hydrogen bonds within itself and with adjacent amino acid residues such as the G588 of the polgamma-A protein. The increase of polymerase activity is very convincingly associated with increase of processivity of the four mutant polgamma-A proteins used in the study (R232H; A467T; W748S; G848S). This is really a remarkable result operated by PZL-A, accompanied by an increase to nearly normal values of the overall replisome function, when the twinkle helicase and mtSSB are added to the in vitro system, together with polgamma-B and mutant polgamma-A species. Likewise, PZL-A has a very remarkable stimulatory effect when exposed to cells with mutations of POLG, pre-treated with ethidium bromide to induce mtDNA depletion. Whilst the untreated mutant cells are virtually unable to recover the normal amount and proliferation rate of mtDNA, the exposure of PZL-A recovers to normal the amount of mtDNA in mutant cells, similar to the wt cells. This is accompanied to a remarkable increase in the oxygen consumption rate in basal and maximal conditions, and ATP production.

Altogether, this is a very accurate, well written and very original investigation on the first compound that appears very effective to stabilize and increase the proficiency of several mutant POLG species responsible of mtDNA depletion in critical organs and devastating diseases in the patients. This work, which is very well conducted in each of its sections, including the statistical analysis, will open the possibility of a rational therapy for one of the most common mitochondrial disorders, due to various mutations in POLG, and possibly help to mitigate the effects on mtDNA metabolism of other conditions, such as mutations in twinkle and mtSSB, or in degenerative disorders in which reduced efficiency of mitochondrial replisome activity may play a role, for instance in several neurodegenerative disorders or aging. This is an important but clearly still preliminary work, since the effects and safety of PZL-A should be tested in vivo, including for instance wt mice or mice with specific

POLG mutations, and eventually patients. It is also unclear which mechanism is implicated in the remarkable effects of PZL-A. The amount of mutant proteins is often reduced in vivo, but the crucial experiments carried out in this paper are in vitro, therefore implying an intrinsic mechanism inducing improved proficiency of the enzyme, despite the presence of well established mutations. So the in-depth mechanism by which PZL-A acts is not clear. Nevertheless, this work is very important because the results collected both in vitro and in vivo are very impressive, and validate the concept that POLG-related disorders, as many others, are hypomorphic and can be mitigated by stabilization and stimulation of the mutant protein. The interaction between the PZL-A specific site and polgamma-B is also interesting because the role of polgamma-B has been repeatedly hypothesized as crucial to improve processivity and increase the efficiency of the replisome system altogether. A role of PZL-A in promoting more efficient interaction between polgamma-B and mutant polgamma-A is a possibility that should be discussed and eventually explored.

The authors are among the best experts in mtDNA replication and expression in normal and disease conditions.

There are possibly some figures that should be re-checked and eventually corrected: for instance in Fig. 4 the ATP production is expressed as $OCR(\text{pmol}/\text{min})/10^4$ cells; in Fig. 2g the amino acid 457 claimed to interact with PZL-A is not indicated.

Referee #2

(Remarks to the Author)

Mammalian mitochondrial DNA (mtDNA) is replicated by the DNA polymerase gamma (POLG) complex, composed of a catalytic POLGA subunit and two accessory POLGB subunits. Over 300 mutations in POLG have been associated with severe, untreatable diseases. This study identifies PZL-A, a novel small molecule that activates mtDNA synthesis by binding to an allosteric site on POLG, unaffected by most disease-causing mutations. PZL-A restores normal activity to three common POLG mutations, improving mtDNA synthesis and cellular respiration in cells from patients with lethal POLG disorders, offering potential for future treatments.

This is a well-executed study, with experiments that are carefully conducted, controlled, and interpreted. The findings have clear and significant clinical implications. The study introduces a novel concept by suggesting that increased processivity of mitochondrial polymerase counteracts defects caused by patient mutations. Overall, the readership of Nature will appreciate the findings. However, there are key points that need to be addressed before publication.

1. It appears that PZL-A stimulates mutant POLG, while the wild-type allele is unaffected, as shown both biochemically and in vivo using mtDNA repopulation assays. Since the hydrophobic pocket to which the compound binds is also present in the wild-type POLG, it is unclear why the WT allele is not overstimulated. It is important to address this in a revised version.
2. In the discussion, the authors suggest that the relevant cells are post-mitotic. However, the in vivo testing was performed on fibroblasts. The effect of the drug should be tested on muscle cells, neurons, or other relevant post-mitotic cell types. Additional functional assays on disease-relevant cell types will provide important insight.
3. In Fig. 4i-k, how does the rescue of respiration and OCR compare to wild-type cells? A comparison would enhance the interpretation of the results.

Referee #3

(Remarks to the Author)

This is an exciting and generally well-written paper that describes the discovery and characterization of a small molecular activator (PZL-A) of the human mitochondrial DNA polymerase that has the potential to overcome heritable mutations that have devastating and deadly effects. The authors provide some kinetic analysis to document the effects of the PZL-A on rates of polymerization for WT and a handful of select mutants, provide cryoEM structural studies to show that the inhibitor binds at the interface between the catalytic (Pol-gamma-A, Polg-A used henceforth) and the accessory protein (Pol-gamma-B, Polg-B), and they offer some cell studies to indicate activity in cells. Overall, the paper is of high interest to a general audience and most of the work is performed and interpreted rigorously, so it will be of interest to wide readership of Nature. I congratulate the authors on their paradigm shifting discovery. However, the paper falls short on the analysis of kinetics and mechanism of action of PZL-A, as detailed below. Because the structural studies appear to be conducted rigorously, I will restrict my comments to the kinetic and equilibrium binding data and their mechanistic interpretation. Comments below are offered in the spirit of desiring to assist the authors to improve and expand an already well-done study.

The pressing question is to understand how the binding of PZL-A at the interface between Polg-A and Polg-B can have such dramatic effects on polymerization rate when there are only small changes seen in the structure. However, the authors seem to be fixated on Polg-B as a "processivity factor." Their efforts to estimate processivity (Fig. 3c) are misguided because this type of experiment is difficult to quantify. Rather, processivity is understood as a ratio of the rates of polymerization (k_{pol}) versus DNA dissociation (k_{off}). Most investigators assume processivity factors have their effect by decreasing DNA dissociation rates. However, the authors have missed important prior work showing that the major effect of the Polg-B "processivity factor" is achieved by increasing the rate of polymerization from 9 to 45 /s, with only a modest change in the DNA dissociation rate from 0.03 /s to 0.02/s. The net effect is to improve processivity from 290 to 2250 nt (Johnson et al. (2000) *Biochemistry* 39:1702-1708). This background provides a more likely postulate for understanding the effect of PZL-A in that it acts by allosterically improving the interaction between Polg-B and Polg-A to stimulate polymerase activity. Although the cryoEM structures do not show measurable changes in structure, a fraction of an Angstrom can produce a 100-

fold increase in rate of an enzyme-catalyzed reaction. Moreover, enzyme dynamics and conformational changes leading to catalysis can involve the whole enzyme structure.

Given this reasoning, it is paramount that measurements of the effects of PZL-A on the rates of polymerization be measured accurately—the lack of quantification of the rates is the weakest part of the paper. For example, key data in Fig. 3a,e and Extended Data Fig. 1 lack quantification. Although for the past three decades it has been common practice for authors to show pictures of gel analysis and to expect the readers to evaluate the changes in band intensity by eye, this is not acceptable. The authors must quantify the time dependence of the reactions to provide a valid estimate of the rate constants governing polymerization for WT and mutant enzyme in the presence and absence of PZL-A. The fundamental data are the heart of the study and would provide a quantitative basis for further structure/activity studies to further improve activity.

There are some rate estimates given in Fig. 1d, but the primary data are not shown, and it is not clear from the writeup whether the rate estimates are derived from a single time point (which is a common source of errors in the literature). For example, a WT enzyme operating at 45 /s will complete synthesis of the 7200 m13 template in less than 2.6 min, which appears by eye to be supported by the data in Extended Data Fig. 1. Thus, showing the time course of the measurements is critical. Another problem is that rates given in Fig. 1d are in nM/min, which is a meaningless number, which requires the reader to dig through the methods section, attempt to ensure that the kinetics experiment described in the methods section (there are several) matches with the one in a given figure, then divide by the enzyme concentration in the experiment to get turnover rates in minutes or seconds. All rate estimates given must be normalized by the enzyme concentration and details of concentrations in the experiments should be provide in the figure legends. Although the table in Fig. 1e does give kcat estimates, values for WT enzyme are not included as a needed reference point.

The template used in Extended Data Fig. 1b was not provided (at least I could not find it). Also, the rates of exonuclease also must be quantified from the data in Extended Data Fig. 1c,d. Without numbers, one cannot support the assertion that the rate of the exonuclease activity was unchanged—unchanged within what range?

A critical part of any enzyme kinetic study is to provide an estimate of the active site concentration (or active fraction) in each enzyme preparation, when possible. The authors provide such data in Extended Data Fig. 5a but appear to misinterpret the results. The authors fit the data to a “non-linear equation” or “one site—specific binding algorithm,” which appears to a hyperbola. However, the data in Fig. 5a deviate from a hyperbola and clearly show tight binding that must be fit using a quadratic equation to afford estimates for both the Kd and the concentration of active enzyme. Fitting the data by eye with a ruler suggests that the binding is very tight (sub nM Kd) but that it requires 10 nM enzyme (nominal concentration) to saturate the binding of 0.7 nM DNA. Ideally, the authors should use these active site titrations to determine the concentrations of active enzyme in their preparations to normalize their kinetic measurements. However, it is unlikely that their enzyme is only 7% active. A more likely explanation is that there are artefacts with the EMSA measurements, which is a common problem. Single turnover polymerization measurements (even with a 3-5 s time point) as a function of DNA concentration is another alternative.

Based on the quantitative analysis suggested in this review, the authors could provide accurate assessment of the rates of polymerization for WT and each of the mutants, and the effects of PZL-A on those rates. Separate estimate of the DNA dissociation rate could then define the effects of processivity. Moreover, the kinetic and structural data could be considered together to provide a reasonable hypothesis for the mechanistic basis for the observed effect of PZL-A on the already established allosteric effects of Polg-B on Polg-A polymerase activity. Moreover, this theory provides a basis to explain the effects of mutations from diverse sites.

One final editorial note: Please do not used the trivial name TWINKLE to describe the human mitochondrial DNA helicase. This is especially troublesome for a paper destined to read by many who are not familiar with jargon pertaining the human mitochondrial DNA replication. How many readers will understand TWINKLE-depend replication?

Version 1:

Reviewer comments:

Referee #1

(Remarks to the Author)

This is the first work to my knowledge that identifies a relatively small pharmacological compound by systematic screening using a quick, yet sophisticated system to analyze a substantial number of compounds. Although future work is warranted to further characterize the compound and eventually produce a derivative suitable for in vivo studies, the results obtained by in vitro and in cell experiments are impressive, very well written, and indeed very encouraging. Several, if not all, my own comments and those of the other reviewing colleagues have been responded to adamantly and satisfactorily. I am confident that this paper will pave the way to start a new era in the so far disappointing field of pharmacological therapy of POLG or other genes involved in mitochondrial DNA replisome.

Referee #2

(Remarks to the Author)

The authors fully addressed my concerns.

Referee #3

(Remarks to the Author)

The summary of key results, originality, etc were covered in my first review.

The authors have thoroughly addressed the concerns of the reviewers. Although one might still quibble about minor points, this is an important paper that describes a landmark discovery and meets the standards of scholarship and importance for publication in Nature. The introduction and discussion provide a good concise summary of the problem and appropriate presentation of the significance and possible underlying mechanisms.

Two minor points:

Lines 79-80: PZL-A increased rates of dNTP incorporation over time. This implies that there is a time-dependent increase in rate of dNTP incorporation, which I don't think the authors have shown. Delete "over time". A rate is the measure of incorporation versus time.

Lines 81-82: "We observed a significant increase of V_{max} and k_{cat} ." V_{max} and k_{cat} measure the same quantity, related by the enzyme concentration. Perhaps the authors meant to say "...increase of measured V_{max} and calculated k_{cat} ." Or they could simply remove reference to V_{max} .

Responses to referees' comments – Nature manuscript 2024-08-17287: "Small-molecule activators restore function of mutant mitochondrial DNA polymerase"

Referee #1 writes: The paper by Valenzuela et al. is a breakthrough piece of work towards the identification of rational, experimentally validated therapeutic approaches in POLG mutations, one of the major causes of mitochondrial disorders in humans. POLG displays a marked heterogeneity in its pathogenic mutations, associated with comparable clinical and pathological diversity.

Our response:

We sincerely thank the reviewer for the positive comments and for recognizing the significance of our work. The valuable feedback provided has encouraged us to extend the discussion and perform additional experiments, which we believe have greatly enhanced the manuscript. Please find detailed responses and revisions below.

Referee #1, comment 1. "The lack of effect on K_m can perhaps be attributed to the fact that the mutations used in the study do not affect directly the catalytic site, but it would be interesting to know the authors' opinion. Likewise, no effect of the PZL-A was recorded for the 3'-5' exonuclease activity of the POLG protein that serves as a proofreading tool during mtDNA replication."

Our response:

We agree with the referee's observations. As pointed out, none of the mutations investigated here affect the catalytic site and we only see minor differences in K_m between wild type and mutant POLY variants. Also, as demonstrated in the new version of the manuscript, PZL-A stabilizes the POLY holoenzyme conformation on DNA, which leads to an increase in k_{cat} and V_{max} , but leaves K_m unchanged.

With respect to exonuclease activity, and as suggested by referee 3, we have in the revised manuscript quantified the data presented in Extended Data Figure 1c-f, which investigate the effects of PZL-A on the exonuclease activity of POLY. Notably, PZL-A appears to enhance the exonuclease activity of certain mutations, making them more similar to the wild-type. This observation aligns with our findings that PZL-A reduces k_{off} , thereby promoting stable interactions with the DNA template. Such stability is crucial for exonuclease activity and highlights the importance of robust template engagement for both polymerase and proofreading functions.

Referee #1, comment 2. This is an important but clearly still preliminary work, since the effects and safety of PZL-A should be tested in vivo, including for instance wt mice or mice with specific POLG mutations, and eventually patients.

Our response:

We would like to point out that the compound used in this study is a tool compound not suitable for in vivo work. However, Pretzel Therapeutics has conducted studies with a closely related compound, which is currently advancing towards clinical phase I trials. The GLP toxicology studies performed to date have not raised any concerns regarding its suitability for future clinical investigations. The phase I study is expected to commence in the spring of 2025.

Additionally, in response to this comment and a point raised by referee #2 (see below), we have expanded our cellular assays to include experiments analyzing the effects of PZL-A on non-dividing fibroblasts carrying POLG mutations to simulate conditions that mimic post-mitotic states, where dNTP levels are significantly reduced (Extended Data Figure 9c-f).

On page 9, last paragraph we now write:

“These experiments were also repeated in non-dividing, quiescent fibroblasts, in which dNTP levels are approximately 10-fold lower, mimicking the conditions in postmitotic tissues (Extended Data Fig. 9c)¹⁹. Notably, PZL-A also stimulated mtDNA repopulation under these conditions (Extended Data Fig. 9d-f).”

We have also conducted experiments using neural stem cells (NSCs) homozygous for the POLG G848S mutation (Extended Data Figure 9m-q). NSCs were chosen because they represent a biologically relevant cell type for mitochondrial diseases, given their role in neurodevelopment and their sensitivity to mitochondrial dysfunction. The results of these additional experiments are included in the revised manuscript, offering further insight into PZL-A’s effects on relevant cell types, including post-mitotic models. We believe these data enhance the relevance of our findings to the context of mitochondrial disease.

On page 10, last paragraph, we now write:

“Finally, we conducted experiments using neural stem cells (NSCs) harboring the severe G848S mutation in homozygous form. NSCs were selected because they represent a biologically relevant cell type for mitochondrial diseases, given their critical role in neurodevelopment and high sensitivity to mitochondrial dysfunction. In these cells as well, PZL-A increased mtDNA levels and improved OXPHOS activity (Extended Data Figure 9m-q).”

Referee #1, comment 3. It is also unclear which mechanism is implicated in the remarkable effects of PZL-A. The amount of mutant proteins is often reduced in vivo, but the crucial experiments carried out in this paper are in vitro, therefore implying an intrinsic mechanism

inducing improved proficiency of the enzyme, despite the presence of well established mutations.

Our reponse:

We agree that certain mutations may lead to a reduction in protein concentrations in vivo. This observation is consistent with the general understanding that protein instability or increased degradation can accompany mutations. However, this does not seem to be the case in the mutant cell lines investigated here. We observed no clear decrease in POL γ A levels in the W748S/R232H or A467T/G848S fibroblasts (Extended Data Figure 9a in the revised version of the manuscript).

On page 9, second paragraph, we now write:

“In the patient cell lines used, the levels of POL γ A were comparable to those observed in wild-type controls (Extended Data Fig. 9a).”

The experiments in this study were performed in vitro, and the data emphasize an intrinsic mechanism by which PZL-A enhances the enzyme's proficiency, independent of potential changes in protein levels observed in vivo. Furthermore, PZL-A demonstrated a positive effect on both stability and activity across the different protein concentrations tested in our experiments (Figure 1c,d, Figures 3a-e, and Extended Data Figure 5a). In addition, our thermal shift experiments demonstrated that PZL-A positively influences POL γ stability (Figure 1f). These findings suggest that PZL-A may exert a beneficial effect even in scenarios where POL γ concentrations are reduced.

Referee #1, comment 4. So the in-depth mechanism by which PZL-A acts is not clear. Nevertheless, this work is very important because the results collected both in vitro and in vivo are very impressive, and validate the concept that POLG-related disorders, as many others, are hypomorphic and can be mitigated by stabilization and stimulation of the mutant protein. The interaction between the PZL-A specific site and polgamma-B is also interesting because the role of polgamma-B has been repeatedly hypothesized as crucial to improve processivity and increase the efficiency of the replisome system altogether. A role of PZL-A in promoting more efficient interaction between polgamma-B and mutant polgamma-A is a possibility that should be discussed and eventually explored.

Our response:

We thank the reviewer for raising this important point. In response to concerns about the mechanism by which PZL-A acts, we have expanded our discussion and conducted additional experiments to provide a clearer understanding of its mode of action. These new data offer a more detailed and comprehensive insight into the mechanism of PZL-A.

In the revised manuscript, we have further investigated the effects of PZL-A on the dissociation rate constant (k_{off}), the dissociation constant (K_d), and the maximal velocity (V_{max}) of polymerase activity. Our results show that PZL-A significantly reduces DNA dissociation rates and enhances polymerase activity. These findings are presented in the updated manuscript and are supported by new figures, including Extended Data Figures 8a-e.

On page 7, last paragraph, we now write:

“In agreement with the observed effect on processivity, we noted that PZL-A stabilizes formation of a complex between POL γ and a primed DNA template. In their stalled, elongating conformations, the mutant forms of POL γ , except W748S, displayed lower affinity (increased K_d) for the template compared to the wild-type enzyme. The addition of PZL-A had a mild stabilizing effect on all mutants (Extended Data Fig. 8a–b).”

We also updated the first paragraph, on page 8, which now read as follows:

“To ensure high replication accuracy, DNA polymerases switch between polymerase and exonuclease modes during active DNA synthesis^{11,13,14}. To investigate if PZL-A enhances the stability of the POL γ -primer-template complex under these dynamic conditions, we initiated DNA synthesis on radioactively labelled primer-template, but omitted two nucleotides, causing POL γ to idle at the primer terminus. Excess cold primer-template (50-fold) was then added, and the stability of the complex (k_{off}) was monitored using an electrophoretic mobility shift assay. The mutant POL γ variants exhibited a higher k_{off} rate than the wild-type, which was significantly reduced upon the addition PZL-A (Extended Data Fig. 8c-e). Taken together, our processivity assay, K_d measurements, and competition experiments demonstrate that PZL-A stabilizes the interaction between mutant POL γ variants and the template during active DNA synthesis.”

While we did not observe clear structural changes upon PZL-A binding, we acknowledge, as noted by referee #3, that even small structural changes at the interface between POL γ A and POL γ B can lead to significant functional effects. In support of this notice, we noted that PZL-A increases POL γ stability in the thermal shift assay. It is also conceivable that PZL-A induces conformational changes that are detectable only during active DNA polymerization, as this process involves transitions between different structural states. Such effects could potentially be observed through single-molecule experiments or ultra-deep cryo-EM structural analysis of the POL γ A-POL γ B complex, capturing transitional states during active DNA replication to uncover conformational changes induced by PZL-A and their role in restoring mutant POL γ function. We plan to explore these possibilities in future studies.

In summary, we believe these new data address the reviewer’s concerns and provide a clearer and more detailed explanation of how PZL-A restores function to mutant POL γ , though some questions remain for future investigation.

Referee #1, comment 5. "There are possibly some figures that should be re-checked and eventually corrected: for instance in Fig. 4 the ATP production is expressed as $\text{OCR}(\text{pmol}/\text{min})/10^4$ cells; in Fig. 2g the amino acid 457 claimed to interact with PZL-A is not indicated."

Our response: We have corrected the labeling in Fig. 4 to accurately reflect ATP production $(\text{pmol}/\text{min})/10^4$ cells. Additionally, we have removed the amino acid interaction in Fig. 2g, as this was an oversight on our part. It was carried over from an earlier version with a different rotamer, and we no longer believe there is an interaction. We have also adjusted the rotamer for L566 and generated new PDB files.

Response to Referee #2:

Referee #2 writes:

Mammalian mitochondrial DNA (mtDNA) is replicated by the DNA polymerase gamma (POLG) complex, composed of a catalytic POLGA subunit and two accessory POLGB subunits. Over 300 mutations in POLG have been associated with severe, untreatable diseases. This study identifies PZL-A, a novel small molecule that activates mtDNA synthesis by binding to an allosteric site on POLG, unaffected by most disease-causing mutations. PZL-A restores normal activity to three common POLG mutations, improving mtDNA synthesis and cellular respiration in cells from patients with lethal POLG disorders, offering potential for future treatments.

This is a well-executed study, with experiments that are carefully conducted, controlled, and interpreted. The findings have clear and significant clinical implications. The study introduces a novel concept by suggesting that increased processivity of mitochondrial polymerase counteracts defects caused by patient mutations. Overall, the readership of Nature will appreciate the findings. However, there are key points that need to be addressed before publication.

Our response:

We thank the reviewer for their thoughtful and constructive comments, as well as for recognizing the significance of our work. The feedback prompted us to conduct additional experiments that we believe have significantly improved the manuscript. We appreciate the opportunity to address these points and strengthen the study. Please see below for details

Referee #2, comment 1. "It appears that PZL-A stimulates mutant POLG, while the wild-type allele is unaffected, as shown both biochemically and in vivo using mtDNA repopulation assays. Since the hydrophobic pocket to which the compound binds is also present in the wild-type POLG, it is unclear why the WT allele is not overstimulated. It is important to address this in a revised version."

PARAGRAPH REDACTED

Referee #2, comment 2. "In the discussion, the authors suggest that the relevant cells are post-mitotic. However, the in vivo testing was performed on fibroblasts. The effect of the drug should be tested on muscle cells, neurons, or other relevant post-mitotic cell types."

Our response:

To address this concern, we have expanded our cellular assays to include experiments analyzing the effects of PZL-A on non-dividing fibroblasts carrying POLG mutations. These experiments were designed to simulate conditions that mimic post-mitotic states, where dNTP levels are significantly reduced (please see Extended Data Figure 9c-f in the revised manuscript). Such assays are well-established in the mitochondrial research field as a proxy for studying post-mitotic cells, given the practical challenges of directly obtaining and working with post-mitotic cells, which would represent a significant undertaking in itself.

In addition, we conducted experiments using neural stem cells (NSCs) with G848S/G848S mutations (please see Extended Data Figure 9m-q in the revised manuscript). NSCs were selected because they represent a biologically relevant cell type for mitochondrial diseases, given their role in neurodevelopment and their sensitivity to mitochondrial dysfunction. These experiments further demonstrate the effects of PZL-A on relevant cell types.

The results of these additional experiments are included in the revised manuscript, providing further insight into PZL-A's effects on both simulated post-mitotic states and biologically relevant models. We believe these data significantly enhance the relevance of our findings in the context of mitochondrial disease.

Please also see our response to Referee #1, comment 2, which addresses the same issue!

Referee #2, comment 3. "In Fig. 4i-k, how does the rescue of respiration and OCR compare to wild-type cells? A comparison would enhance the interpretation of the results."

Our response:

As recommended, we have added comparisons to wild-type cells in Fig. 4i-k to enhance the interpretation of the respiration and OCR rescue data.

Response to Referee #3:

Referee #3 writes:

This is an exciting and generally well-written paper that describes the discovery and characterization of a small molecular activator (PZL-A) of the human mitochondrial DNA polymerase that has the potential to overcome heritable mutations that have devastating and deadly effects. The authors provide some kinetic analysis to document the effects of the PZL-A on rates of polymerization for WT and a handful of select mutants, provide cryoEM structural studies to show that the inhibitor binds at the interface between the catalytic (Pol-gamma-A, Polg-A used henceforth) and the accessory protein (Pol-gamma-B, Polg-B), and they offer some cell studies to indicate activity in cells. Overall, the paper is of high interest to a general audience and most of the work is performed and interpreted rigorously, so it will be of interest to wide readership of Nature. I congratulate the authors on their paradigm shifting discovery. However, the paper falls short on the analysis of kinetics and mechanism of action of PZL-A, as detailed below. Because the structural studies appear to be conducted rigorously, I will restrict my comments to the kinetic and equilibrium binding data and their mechanistic interpretation. Comments below are offered in the spirit of desiring to assist the authors to improve and expand an already well-done study.

Our response:

We thank the reviewer for their thoughtful and constructive comments, as well as for recognizing the significance of our work. The feedback on the kinetic and mechanistic analysis of PZL-A was particularly helpful and prompted us to conduct additional experiments that we believe have significantly improved the manuscript. We appreciate the opportunity to address these points and strengthen the study. Please see below for details.

Referee #3, comment 1.

The pressing question is to understand how the binding of PZL-A at the interface between Polg-A and Polg-B can have such dramatic effects on polymerization rate when there are only small changes seen in the structure. However, the authors seem to be fixated on Polg-B as a “processivity factor.” Their efforts to estimate processivity (Fig. 3c) are misguided because this type of experiment is difficult to quantify.

Rather, processivity is understood as a ratio of the rates of polymerization (k_{pol}) versus DNA dissociation (k_{off}). Most investigators assume processivity factors have their effect by decreasing DNA dissociation rates. However, the authors have missed important prior work showing that the major effect of the Polg-B “processivity factor” is achieved by increasing the rate of polymerization from 9 to 45 /s, with only a modest change in the DNA dissociation rate from 0.03 /s to 0.02/s. The net effect is to improve processivity from 290 to 2250 nt (Johnson et al. (2000) *Biochemistry* 39:1702-1708).

Our response:

We thank the reviewer for the valuable and constructive comments. We have performed a series of new experiments to address the reviewer's concerns (see below). As noted, the processivity assay is indeed challenging to quantify, even though it provides a striking illustration of how POL γ mutations affect processivity and how PZL-A can overcome this effect.

Prompted by the reviewer's suggestions, we have re-analyzed the effects of PZL-A on K_d and k_{off} for interactions with a primed DNA template. Additionally, we have determined the active concentration of POL γ (Extended Data Figure 1b) and recalculated k_{cat} .

K_d measurements: We repeated K_d measurements with POL γ B in excess to stabilize the heterotrimer, particularly at lower POL γ A concentrations. Under these conditions, PZL-A stabilizes interactions with a primed template and causes a mild decrease in the apparent K_d . Please see Extended Data Figure 8a,b in the revised manuscript.

k_{off} analysis: We analyzed the effects of PZL-A on k_{off} for all mutant variants of POL γ and compared these with the k_{off} of wild-type POL γ . The results show that k_{off} decreases significantly for all mutants. Please see Extended Data Figure 8c-e in the revised manuscript.

On page 7, last paragraph, we now write:

“In agreement with the observed effect on processivity, we noted that PZL-A stabilizes formation of a complex between POL γ and a primed DNA template. In their stalled, elongating conformations, the mutant forms of POL γ , except W748S, displayed lower affinity (increased K_d) for the template compared to the wild-type enzyme. The addition of PZL-A had a mild stabilizing effect on all mutants (Extended Data Fig. 8a–b).”

We have also updated the first paragraph, on page 8, which now read as follows:

“To ensure high replication accuracy, DNA polymerases switch between polymerase and exonuclease modes during active DNA synthesis^{11,13,14}. To investigate if PZL-A enhances the stability of the POL γ -primer-template complex under these dynamic conditions, we initiated DNA synthesis on radioactively labelled primer-template, but omitted two nucleotides, causing POL γ to idle at the primer terminus. Excess cold primer-template (50-fold) was then added, and the stability of the complex (k_{off}) was monitored using an electrophoretic mobility shift assay. The mutant POL γ variants exhibited a higher k_{off} rate than the wild-type, which was significantly reduced upon the addition PZL-A (Extended Data Fig. 8c-e). Taken together, our processivity assay, K_d measurements, and competition experiments demonstrate that PZL-A stabilizes the interaction between mutant POL γ variants and the template during active DNA synthesis.”

Active POL γ concentrations and recalculated k_{cat} : Using a primed DNA template in a gel-shift assay, we determined the active concentrations of the POL γ holoenzyme (Extended Data Fig 1b). These values allowed us to recalculate k_{cat} . The results indicate a clear increase in k_{cat} in the presence of PZL-A. Please see Figure 1d,e in the revised manuscript.

We acknowledge that the previous version of the manuscript placed excessive focus on processivity. The new data demonstrate that there is a combined effect on both polymerization and dissociation, as suggested by the reviewer.

In the revised version of the discussion, on page 11, we now write:

“Specifically, PZL-A binds to an allosteric site formed at the interface between POL γ A and POL γ B, which is unaffected by the most common disease-causing mutations². Binding of PZL-A increases the stability of POL γ bound to template DNA, evidenced by a reduction in K_d and k_{off} , and enhances the k_{cat} of the enzyme.”

Later in the same paragraph, we add:

“This explains why PZL-A, which enhances POL γ -DNA binding stability and increases the enzyme’s catalytic turnover, can broadly restore POL γ function across different pathogenic mutations.”

Please note that the precise numbers for nucleotide incorporation rates reported here cannot be directly compared to those obtained by Johnson et al. 2000, since they measured the single nucleotide incorporation rate under non-limiting template concentrations, whereas we assessed the formation of long DNA products in the presence of mtSSB and with limiting amounts of template. Johnson et al. estimated the maximum DNA polymerization rate as $45 \pm 1 \text{ s}^{-1}$. Our maximum velocity was approximately 3 nts s^{-1} , which closely aligns with the polymerization rate observed in vivo (Clayton et al., 1982, Mitochondrial DNA replication and its regulation) and in experiments with the complete mtDNA replisome in vitro (Korhonen et al., 2004, Reconstitution of a minimal mtDNA replisome in vitro). Despite these methodological differences, the critical observation from our work is the substantial impact of PZL-A on incorporation rates across all tested mutants, highlighting its potential to enhance POL γ function under conditions that approximate physiological relevance. We have included a reference to the Johnson et al paper in the new version of the manuscript.

Referee #3, comment 2. This background provides (referring to the previous comment from reviewer 3) a more likely postulate for understanding the effect of PZL-A in that it acts by allosterically improving the interaction between Polg-B and Polg-A to stimulate polymerase activity. Although the cryoEM structures do not show measurable changes in structure, a fraction of an Angstrom can produce a 100-fold increase in rate of an enzyme-catalyzed reaction. Moreover, enzyme dynamics and conformational changes leading to catalysis can involve the whole enzyme structure.

Our response:

We fully agree with the reviewer that even small structural changes at the interface between POL γ A and POL γ B can result in significant functional effects. Indeed, when we analyze V_{max} , K_d , k_{cat} and k_{off} , we do see an obvious effect, as discussed in our response to the reviewer's first comment (see above).

While we did not observe clear structural changes upon PZL-A binding, we acknowledge that even small structural changes at the interface between POL γ A and POL γ B can lead to significant functional effects. In support of this notice, we noted that PZL-A increases POL γ stability in the thermal shift assay. It is also conceivable that PZL-A induces conformational changes that are detectable only during active DNA polymerization, as this process involves transitions between different structural states. Such effects could potentially be observed through single-molecule experiments or ultra-deep cryo-EM structural analysis of the POL γ A-POL γ B complex, capturing transitional states during active DNA replication to uncover conformational changes induced by PZL-A and their role in restoring mutant POL γ function. We plan to explore these possibilities in future studies.

Referee #3, comment 3.

Given this reasoning, it is paramount that measurements of the effects of PZL-A on the rates of polymerization be measured accurately—the lack of quantification of the rates is the weakest part of the paper. For example, key data in Fig.3a,e and Extended Data Fig. 1 lack quantification. Although for the past three decades it has been common practice for authors to show pictures of gel analysis and to expect the readers to evaluate the changes in band intensity by eye, this is not acceptable. The authors must quantify the time dependence of the reactions to provide a valid estimate of the rate constants governing polymerization for WT and mutant enzyme in the presence and absence of PZL-A. The fundamental data are the heart of the study and would provide a quantitative basis for further structure/activity studies to further improve activity.

Our response:

We thank the reviewer for the insightful comments and fully agree that accurate quantification is paramount to ensuring the robustness and reproducibility of our findings. We have carefully addressed the reviewer's concerns and revised the manuscript to include quantitative analyses where they were previously missing.

In response to the request for quantification, we have now included statistical measurements of PZL-A stimulation of DNA polymerase activity (Figure 1c) and kinetics (Figure 1d). These measurements were repeated three times, and we now report statistics.

The thermal shift experiments mentioned in Figure 1f, demonstrating interactions between PZL-A and POL γ , have also been repeated three times, and we have calculated the standard deviations, which are indicated in the figure.

While Figure 3e is intended as a visual representation, we acknowledge the need for quantitative data to complement this illustration. We now explicitly state in the figure legend that the experiment was repeated three times for reproducibility, and we have also quantified the data. Please see Extended Data Figure 8f.

We have also quantified the data presented in Extended Data Figure 1c-f, which investigate effects of PZL-A on the exonuclease activity of POL γ . If anything, PZL-A improves the exonuclease activity of some mutations, whereas WT POL γ remains unaffected.

We have also performed new measurements of k_{off} in Extended Data Figure 8c-e. These experiments have been performed three times and standard deviations are indicated.

We have also reanalyzed K_d , as discussed above. The measurements were performed three times and the standard deviations are indicated (Extended Data Figure 8a,b).

Two of the experiments (Figure 3a and Extended Data Figure 1a) were only performed to provide a visual illustration of the effects of PZL-A on DNA synthesis. These two experiments are performed using a 5'-labeled primer that are elongated by POL γ . The experimental setup means that the signal remains unchanged during the experiment and only the sizes of the products increase, which makes quantification challenging.

In Figure 1d, the effects of PZL-A were analyzed in time-course experiments performed at eight different dNTP levels at low protein concentration to ensure the experiment was performed within the linear range (each experiment was performed at least three times). These experiments provided the data used to calculate V_{max} , K_m , and k_{cat} . To visually illustrate these effects, we also include Figure 3a and Extended Data Figure 1a. Figure 3a shows a time-course experiment at a single fixed dNTP concentration, while Extended Data Figure 1a presents a fixed-time experiment at increasing dNTP levels. Although the data in Figure 3a and Extended Data Figure 1a were also repeated three times, these results are not suitable for further quantification. At later time points (Figure 3a) or at higher dNTP concentrations (Extended Data Figure 1a), the reactions result in the formation of full-length replication products, which can no longer accumulate in a linear manner. These experiments are also performed at higher protein concentrations than Figure 1d. However, these experiments are included to visually demonstrate the variation in product size, which provides complementary information to the quantitative analysis in Figure 1d.

We wish to highlight that all experiments, including those shown in these figures, were repeated at least three times. This is now explicitly stated in the manuscript. Furthermore, all repeat data for these and other experiments are included in the source data section.

Referee #3, comment 4. There are some rate estimates given in Fig. 1d, but the primary data are not shown, and it is not clear from the writeup whether the rate estimates are derived from a single time point (which is a common source of errors in the literature). For example, a WT enzyme operating at 45 /s will complete synthesis of the 7200 m13 template in less than 2.6 min,

which appears by eye to be supported by the data in Extended Data Fig. 1. Thus, showing the time course of the measurements is critical.

Our response:

We are grateful for the opportunity to clarify these points. We agree with the reviewer that single time-point measurements can introduce error, and we should have explicitly stated that the data in Figure 1d were derived from multiple time points. Measurements were taken at 0, 2.5, 5, 7.5, 10, and 15 minutes at each indicated dNTP concentration. These values were then used to calculate V_{max} , K_m , and k_{cat} . For source data, see Supplementary Figs. 1–3.

Regarding Extended Data Figure 1, we believe there may be a misunderstanding. This is not a time-course experiment but rather shows the dependence of DNA synthesis on varying dNTP concentrations. The experiment was performed at a fixed time point using a 5'-labeled primer.

Referee #3, comment 5. Another problem is that rates given in Fig. 1d are in nM/min, which is a meaningless number, which requires the reader to dig through the methods section, attempt to ensure that the kinetics experiment described in the methods section (there are several) matches with the one in a given figure, then divide by the enzyme concentration in the experiment to get turnover rates in minutes or seconds. All rate estimates given must be normalized by the enzyme concentration and details of concentrations in the experiments should be provide in the figure legends. Although the table in Fig.1e does give k_{cat} estimates, values for WT enzyme are not included as a needed reference point.

Our response:

We have followed the reviewer's suggestions. As discussed above, we have calculated the active concentrations of the POL γ holoenzyme and normalized all rate estimates by enzyme concentration to provide turnover rates (k_{cat}) in Figure 1e. Active enzyme concentration details have been included in the Extended Data Figure 1b, which is now referenced in the figure legends of Figure 1e. Additionally, the k_{cat} for WT POL γ has been added to Fig. 1e as a reference. Figure legends and the methods section have been clarified to improve interpretation.

Referee #3, comment 6.

The template used in Extended Data Fig. 1b was not provided (at least I could not find it).

Our response:

The template used in Extended Data Fig. 1b (1c in the new versions) is now presented in the methods' section, under the subheading "Exonuclease activity".

Referee #3, comment 7.

Also, the rates of exonuclease also must be quantified from the data in Extended Data Fig. 1c,d. Without numbers, one cannot support the assertion that the rate of the exonuclease activity was unchanged—unchanged within what range?

Our response:

We have now quantified the data presented in Extended Data Figure 1d-f in the revised manuscript, which examine the effects of PZL-A on the exonuclease activity of POL γ . The results suggest that, if anything, PZL-A enhances the exonuclease activity of certain mutations, while wild-type POL γ remains unaffected.

Referee #3, comment 8.

A critical part of any enzyme kinetic study is to provide an estimate of the active site concentration (or active fraction) in each enzyme preparation, when possible. The authors provide such data in Extended Data Fig. 5a but appear to misinterpret the results. The authors fit the data to a “non-linear equation” or “one site—specific binding algorithm,” which appears to a hyperbola. However, the data in Fig. 5a deviate from a hyperbola and clearly show tight binding that must be fit using a quadratic equation to afford estimates for both the K_d and the concentration of active enzyme. Fitting the data by eye with a ruler suggests that the binding is very tight (sub nM K_d) but that it requires 10 nM enzyme (nominal concentration) to saturate the binding of 0.7 nM DNA. Ideally, the authors should use these active site titrations to determine the concentrations of active enzyme in their preparations to normalize their kinetic measurements. However, it is unlikely that their enzyme is only 7% active. A more likely explanation is that there are artefacts with the EMSA measurements, which is a common problem. Single turnover polymerization measurements (even with a 3-5 s time point) as a function of DNA concentration is another alternative.

Based on the quantitative analysis suggested in this review, the authors could provide accurate assessment of the rates of polymerization for WT and each of the mutants, and the effects of PZL-A on those rates. Separate estimate of the DNA dissociation rate could then define the effects of processivity. Moreover, the kinetic and structural data could be considered together to provide a reasonable hypothesis for the mechanistic basis for the observed effect of PZL-A on the already established allosteric effects of Polg-B on Polg-A polymerase activity. Moreover, this theory provides a basis to explain the effects of mutations from diverse sites.

Our response:

Thank you for your detailed and constructive feedback. As suggested, we calculated the active concentrations of the POL γ holoenzyme using a primed DNA template in a gel shift assay (Extended Data Figure 1b). Using these values, we recalculated k_{cat} , and the data reveal that PZL-A increases both V_{max} and k_{cat} . Please refer to Figure 1e in the revised manuscript.

We also repeated K_d measurements with POL γ B in excess to stabilize the heterotrimer at lower POL γ A concentrations. Please see the new Extended Data Figure 8a,b in the manuscript. Additionally, we analyzed the data using a quadratic equation, as recommended. This approach allowed us to estimate K_d more accurately, confirming that PZL-A stabilizes interactions with a primed template and decreases the apparent K_d .

While we did not perform single-turnover polymerization measurements, we agree that such experiments, along with single-molecule studies, would provide deeper insights. We have initiated work on single-molecule measurements and plan to conduct more detailed enzyme kinetic analyses in future studies. However, we believe these experiments extend beyond the scope of the current manuscript and would be better addressed in a subsequent paper.

Referee #3, comment 9.

One final editorial note: Please do not use the trivial name TWINKLE to describe the human mitochondrial DNA helicase. This is especially troublesome for a paper destined to read by many who are not familiar with jargon pertaining to the human mitochondrial DNA replication. How many readers will understand TWINKLE-dependent replication?

Our response:

We have followed the suggestion and replaced “TWINKLE” with “mitochondrial DNA helicase” throughout the manuscript. In instances where the name TWINKLE is used, we have ensured it is combined with “DNA helicase,” i.e., “TWINKLE DNA helicase,” to enhance clarity and make the text more accessible to readers unfamiliar with the terminology.